# Engineering consortia by polymeric microbial swarmbots

Lin Wang[1], Xi Zhang[1], Chenwang Tang[1], Pengcheng Li[1], Runtao Zhu[1], Jing Sun[2], Yunfeng Zhang[1], Hua Cui[1], Jiajia Ma[3], Xinyu Song[3], Weiwen Zhang[3], Xiang Gao[1], Xiaozhou Luo[1], Lingchong You [4], Ye Chen [1] & Zhuojun Dai [1✉]

Synthetic microbial consortia represent a new frontier for synthetic biology given that they can solve more complex problems than monocultures. However, most attempts to co-cultivate these artificial communities fail because of the winner-takes-all in nutrients competition. In soil, multiple species can coexist with a spatial organization. Inspired by nature, here we show that an engineered spatial segregation method can assemble stable consortia with both flexibility and precision. We create microbial swarmbot consortia (MSBC) by encapsulating subpopulations with polymeric microcapsules. The crosslinked structure of microcapsules fences microbes, but allows the transport of small molecules and proteins. MSBC method enables the assembly of various synthetic communities and the precise control over the subpopulations. These capabilities can readily modulate the division of labor and communication. Our work integrates the synthetic biology and material science to offer insights into consortia assembly and serve as foundation to diverse applications from bio-manufacturing to engineered photosynthesis.

[1] CAS Key Laboratory of Quantitative Engineering Biology, Center for Materials Synthetic Biology, Guangdong Provincial Key Laboratory of Synthetic Genomics, Shenzhen Institute of Synthetic Biology, Shenzhen Institute of Advanced Technology, Chinese Academy of Sciences, Shenzhen 518055, China. [2] Soft Bio-interface Electronics Lab, Center of Neural Engineering, CAS Key Laboratory of Human-Machine Intelligence-Synergy Systems, Shenzhen Institute of Artificial Intelligence and Robotics for Society, Shenzhen Institutes of Advanced Technology, Chinese Academy of Sciences, Shenzhen 518055, China. [3] Laboratory of Synthetic Microbiology, School of Chemical Engineering & Technology, Tianjin University, Tianjin 300072, China. [4] Department of Biomedical Engineering, Duke University, Durham, NC 27708, USA. ✉email: zj.dai@siat.ac.cn

Synthetic biology has gained increasing interests in engineering microbial consortia since they have unique and attractive features not shared by individual populations[1–4]. For example, members of the consortia can communicate by detecting and responding to one another through signal molecules exchange[5,6]. Subpopulations can execute different tasks by division of labor, such that the whole community can perform complex functions that cannot be achieved by single-strain systems[7,8]. In nature, microbial consortia play essential roles in multiple aspects. For example, microbial communities in the soil can fix nitrogen and decompose organic matter in the carbon cycle[9]. Gut microbiome plays a critical role in metabolizing nutrients and preventing pathogen invasion[10,11]. Therefore, the capability of synthesizing functional microbial consortia can potentially benefit diverse areas, including biomanufacturing, biomedicine, and bioremediation.

Despite the obvious advantages, harnessing the functions of consortia is much more difficult than monocultures in homogeneous laboratory condition. Most attempts to co-cultivate the artificial microbial communities fail mostly due to the mismatched rates in the consumption and production of nutrients among the subpopulations[12–14]. Recently, there are examples of programming the synthetic microbial consortia to produce the multiprotein system. Nevertheless, members of these communities are mostly within the same species (e.g., E. coli) and even the same strains[15,16]. Efforts have also been taken in constructing a mutualistic relationship between E. coli and S. cerevisiae for co-cultivation. However, this strategy is uniquely designed for the paired microorganisms and is not generalizable[17]. Besides, there is a lack of methods to precisely control the ratio across the species, in spite that the composition can significantly affect both the communication and the division of labor between the subpopulations[13,18].

Natural microbial communities maintain their structural stability through various strategies such as mutualism and symbiosis. By far, most synthetic microbial consortia have been designed using pairwise interactions[7,19–21]. However, as a microbial community increases in the number of different populations, the community complexity increases due to the combinatorial increase in the number of pairwise interactions and the emergence of higher-order interactions[22]. The complexity of controlling these interactions makes constructing and predicting the dynamics of these networks difficult, even in small communities[3,23]. In nature, another commonly used strategy by microorganisms to achieve the balance of the community is to inhabit with a spatial organization. For example, species of bacteria in soil coexist in the form of microcolonies with a few hundred micrometers distance[24]. This spatial organization has been hypothesized to be a determinant in maintaining a stable microbial ecosystem[12,25–27].

Inspired by nature, here we develop a microcapsule-based spatial arrangement method to construct communities made of single or multispecies. We engineer a microbial swarmbot (MSB) consisting of a polymeric microcapsule encapsulating the engineered bacteria (Fig. 1). The microcapsule has a three-dimensional cross-linked structure, which allows the transport of proteins and small molecules including nutrients, signaling molecules, and metabolites, but traps the microbes[28]. Thus, each MSB represents one subcommunity that can interact with other MSBs. We first generate individual MSB by encapsulating the subpopulations, and then co-culture these MSBs to form the microbial swarmbot consortia (MSBC). Polymeric capsules insulate populations in different MSB. Therefore, members of the consortium, even with a mismatched growth rate, can grow in balance due to a relevantly independent growth space and defined carrying capacity. Utilizing this strategy, we assembled a single-species E. coli consortium comprising 34 strains, two-species consortia containing E. coli and S. cerevisiae

or E. coli and P. pastoris, a multispecies consortium containing E. coli, S. cerevisiae and C. glutamicum, and a phototrophic microbial consortium containing S. elongatus and E. coli. Notably, MSBC platform enables precise control over the subpopulations. By exploiting this feature, we are able to modulate the communication and division of labor even across the species.

## Results

**Engineering MSB of different microorganisms**. We used the naturally occurring polysaccharide chitosan as the encapsulating material. The resulting three-dimensional cross-linked networks trapped the living cells but permitted the delivery of small molecules and proteins (Supplementary Fig. 1). We used chitosan capsules to encapsulate the mCherry-expressing E. coli or GFP-expressing S. cerevisiae. Our results showed that both E. coli and S. cerevisiae grew and expressed the fluorescent proteins, suggesting that the nutrients could pass through the MSB and support the cell growth (Fig. 2a, b). The porous structure of capsules can naturally separate the cell factory and the biomacromolecules. We encapsulated a model polymer (rhodamine-conjugated dextran, Mw = 150 kDa) inside the capsule and measured its releasing profile. Results showed that the polymer diffused gradually to the supernatant (Supplementary Fig. 2). These capabilities of MSB could be coupled with the engineered auto-lysing E. coli or secretory P. pastoris for protein manufacturing. Compared with the conventional method, use of MSB integrated the cell lysis and product separation, which largely facilitated the downstream process[28]. We first programmed the E. coli with a genetic circuit that led to the density-dependent autolysis of the cells and generated the MSB (Supplementary Fig. 3). The purified supernatant of the MSB culture revealed the successful production and transport of the model proteins (polypeptides tagged GFP) (Fig. 2c). Next, we fused the sequence of a therapeutic protein (recombinant human growth hormone, rh-GH) with α-factor signal for secretory expression (driven by the AOXI promoter). The circuit was then linearized and integrated into the genome of P. pastoris strain X-33. The resultant MSB(P. pastoris) produced and transported the rh-GH into the supernatant, as verified by the SDS-PAGE results (Fig. 2d).

Besides biomacromolecules, MSB system could also produce valuable small molecules such as cannabigerolic acid (CBGA) by simply encapsulating a metabolically engineered microbial host. In nature, CBGA is produced from olivetolic acid (OA) and the mevalonate-pathway derived intermediate geranylpyrophosphate (GPP) catalyzed by a geranylpyrophosphate:olivetolate geranyltransferase (GOT). Therefore, we engineered a GPP-overproducing and GOT-expressing S. cerevisiae (yCAN14)[29], encapsulated the cells and cultured them in the medium with 1 mM OA. HPLC and LC/MS results confirmed that the substrate (OA) was delivered through the porous capsules and converted to the CBGA by the MSB(S. cerevisiae) after 96 h of fermentation (Fig. 2e and Supplementary Figs. 4 and 5). The yield of CBGA by MSB(S. cerevisiae) was comparable with that of the free cells (Supplementary Fig. 6), underscoring the metabolic activity of the cells inside MSB.

**MSBC strategy enables the assembly of 34-strain consortia**. After verifying the functions of the MSB, we next assembled a 34-strain microbial swarmbot consortium (MSBC) for multi-proteins manufacturing. One of the most attractive features of natural or engineered microbial consortia is the division of labor: the subpopulations perform a combination of tasks that are difficult or potentially impossible for single strain due to the metabolic burden. This characteristic is particularly powerful for preparing multiprotein systems, such as the 34-protein involved PURE (protein synthesis using recombinant elements) machinery

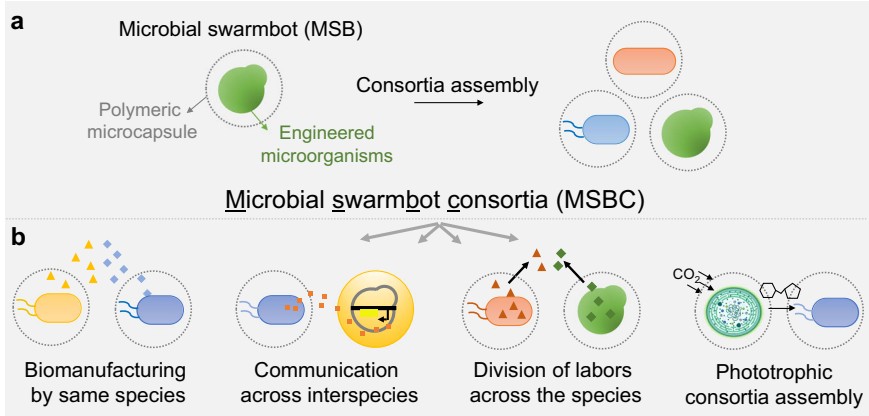

**Fig. 1 Engineering microbial consortia with polymeric microbial swarmbots. a** Engineering stably coexisting consortia by microbial swarmbot mediated spatial segregation strategy. Each microbial swarmbot (MSB) consists of a small population of engineered microbes encapsulated in polymeric microcapsules. The MSBs containing different subpopulations are then co-cultured to form the microbial swarmbot consortia (MSBC). Polymeric capsules insulate populations in different MSB. Therefore, members of the consortium, even with a mismatched growth rate, can grow in balance due to a relevantly independent growth space and defined carrying capacity of the microcapsules. **b** MSBC strategy enables diverse applications. MSBC method can construct versatile consortia for diverse applications. For example, MSBC (single species) can collectively manufacture multienzyme system. Members of the MSBC (across the species) can communicate or collaborate (division of labor). Phototrophic MSBC can transform the carbon dioxide into the carbon source by phototrophic members to maintain the growth of the heterotrophic members.

(Fig. 3a)[30]. To implement the assembly, we created a set of 34 MSBs, each encapsulating a strain carrying an autolysis circuit (Supplementary Fig. 3), and a plasmid that encodes the expression of one PURE component. Individually cultured MSB could produce each of the 34 proteins (Fig. 3b). We then created the MSBC by mixing the 34 MSBs, cultured MSBC in M9 medium, collected, and purified the supernatant (all enzymes are His-tagged). Mass spectrometry data confirmed the presence of all essential elements (Fig. 3c). The reconstitution of in vitro protein expression system including the 34 enzymes (product of MSBC) and other essential components collectively led to the expression of RFP, underscoring the activities of all enzymes produced by 34-strain MSBC (Fig. 3d). Compared with the traditional preparing methods by monocultures or homogeneous consortia which need repeated culturing and downstream processing steps, MSBC platform inherited the characteristics of MSB platform, and integrated the multiple steps of production, disruption and separation into a concise format[28,31]. A functional multiprotein cocktail can be produced by MSBC with limited or no access to centrifuge, sonicator or electricity, and the operation is simple and can be conducted after minimal training (Supplementary Fig. 7).

**MSBC strategy enables the assembly of multispecies consortia with precise population control.** Compared with the consortia made of the same species, there are significantly more complicated interactions in the communities of multispecies[14]. In the absence of a stabilizing mechanism, co-culturing these consortia often results in the domination of the faster-growing population. For example, *E. coli* and *S. cerevisiae* are two of the most widely studied and exploited microbes in both laboratory and industry. Co-culturing the *E. coli* and *S. cerevisiae* can potentially harness the pros of the two systems, such as the high-yield protein production of *E. coli* and soluble expression of sophisticated eukaryotic enzymes in *S. cerevisiae*. Nevertheless, the difference in the doubling time of *E. coli* (~20 min) and *S. cerevisiae* (~90–120 min) makes the *E. coli* more effective in consuming the nutrients and thus becoming dominant. To verify this, we inoculated the *E. coli* (expressing mCherry) and *S. cerevisiae* (expressing GFP) into the medium (SC + LB, see "Methods"),

cultured the well-mixed consortium for 24 h and analyzed the composition by flow cytometer. We created a series of homogeneous consortia by altering the initial seeding ratio between *E. coli* and *S. cerevisiae* (keeping the same total cell number). In line with our prediction, results showed that in all occasions, *E. coli* persisted in the communities at the end regardless of the composition control at the initial stage (Fig. 4a and Supplementary Fig. 8). We next examined the performance of MSBC strategy in assembling the same consortium. We generated the MSB(*E. coli*) and MSB(*S. cerevisiae*), co-cultured these two MSBs (with different seeding ratios) to create multiple MSBC. Results showed that the MSBC platform successfully enabled the co-culturing of the two species, as evidenced by the proliferation of both species inside their own niche without interference (Fig. 4b and Supplementary Movie 1). Since each MSB has a defined carrying capacity, the composition of the consortia could be precisely modulated by adjusting the seeding ratio of two species (MSB) (Fig. 4c).

Versatile consortia can be built by MSBC platform with flexibility. In this notion, we generated three different MSBs containing *E. coli* (BFP tagged), *S. cerevisiae* (GFP tagged) or *C. glutamicum* (mCherry tagged), and verified the growth of these MSBs using a micro-device with three independent chambers (Fig. 4d and Supplementary Fig. 9). We then constructed various MSBC, including three two-species consortia (chamber 1–3, 1:1 in seeding ratio) and a three-species consortium (chamber 4, 1:1:1 in seeding ratio) in a micro-device with four connected culturing chambers (Fig. 4e). Again, our results confirmed that the subpopulations of MSBC grew actively with the clear boundaries insulated by the cross-linked polymer. In comparison, homogeneous culture of the *E. coli*, *S. cerevisiae* and *C. glutamicum* (seeding ratio = 1:1:1) led to the domination of *E. coli* at the end (Supplementary Fig. 10a). By simply switching the different MSB, MSBC strategy can plug-and-play to create an array of combinations with both flexibility and precision (Fig. 4f). The patterns of MSBC could last for over 7 days, underscoring the stability of the system (Supplementary Fig. 10b).

**MSBC strategy can modulate DOL and communication across the species.** The attractive traits of microbial consortia rely on

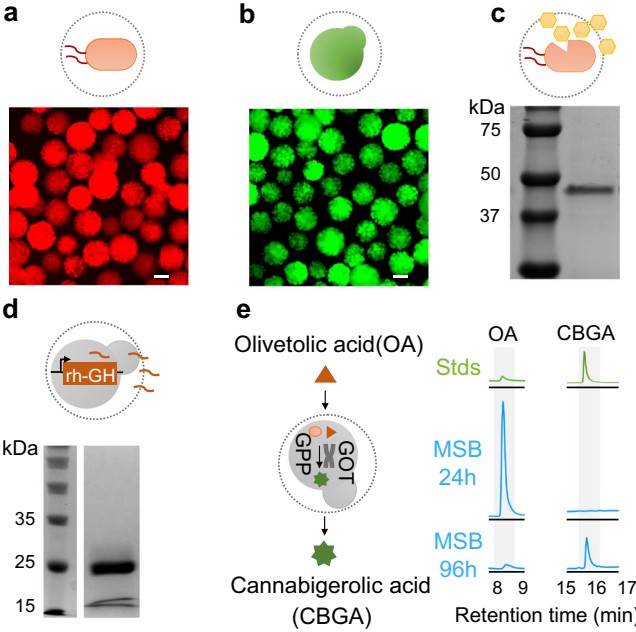

**Fig. 2 Microbial swarmbot supported the growth of various microbes and produced versatile proteins and small molecules. a** The growth of engineered *E. coli* was supported in microbial swarmbot (MSB). MG1655 cells (constitutively producing mCherry) were encapsulated in chitosan capsules and cultured in the SC + LB medium at 30 °C. Images were taken at 24 h. Scale bar = 200 μm. **b** The growth of engineered *S. cerevisiae* was supported in MSB. BY4742 cells (expressing GFP under the TDH3 promoter) were encapsulated in chitosan capsules and cultured in the SC + LB medium at 30 °C. Images were taken at 24 h. Scale bar = 200 μm. **c** MSB(*E. coli*) produced the model protein through the programmed self-lysis of the cells. We used a density-dependent circuit (ePop) to program the autonomous lysis of the *E. coli*. MSB(*E. coli*) (BL21(DE3) carrying the ePop circuit and the GFP-fused polypeptides expression circuit) were cultured for 24 h in M9 medium. The supernatant was collected for His-tagged resins purification. The resultant protein was verified by SDS-PAGE. Experiments were repeated independently more than three times with similar results. **d** MSB(*P. pastoris*) produced a therapeutic protein through the engineered secretion. We created the circuit by fusing the sequence of recombinant human growth hormone (rh-GH) with α-factor signal for secretory expression. The circuit (driven by the AOX1 promoter) was then linearized and integrated into the genome of *P. pastoris* X-33 by electroporation. We generated the MSB(*P. pastoris*) and cultured them in the BMMY medium for 96 h. The supernatant was then collected, purified by His-tagged resins and verified by SDS-PAGE. Experiments were repeated independently more than three times with similar results. **e** MSB(*S. cerevisiae*) produced cannabigerolic acid (CBGA) by feeding olivetolic acid (OA). MSB (*S. cerevisiae*) were cultured in 1 mM olivetolic acid and assayed for CBGA production using liquid chromatography. The decrease in the OA peak and increase in the CBGA peak indicated the production of CBGA by consuming OA. Source data are provided as a Source Data file.

multiple unique features. Besides division of labor (DOL), members of the consortium can communicate with one another by trading metabolites or by exchanging dedicated molecular signals. To examine these two features, we first built a MSBC of *E. coli* and *P. pastoris* to generate DOL product that could collectively degrade the antibiotics and organophosphates (often present in the agriculture wastewater due to the overuse of veterinary medicines and pesticides). In particular, we engineered the MSB(*E. coli*) to produce the β-lactamase (Bla) since it could degrade a board class of β-lactams antibiotics by breaking β-lactam ring[32,33]; and MSB(*P. pastoris*) to produce the recombinant human paraoxonase 1 (rh-PON1)

since it could detoxify organophosphate insecticides such as paraoxon, parathion and chlorpyrifos by hydrolysis (Fig. 5a)[34,35]. Culturing the resultant MSBC led to the production of both enzymes (Fig. 5b and Supplementary Fig. 11). We used a cell-rescue assay to evaluate the antibiotics degrading capability of DOL product (schematic in Fig. 5c). Briefly, LB medium supplemented with ampicillin was aliquoted in a 96-well plate. After adding the DOL product, cells (sensitive to antibiotics) were immediately inoculated and the growth of the cells was monitored by a platereader. Results validated that with the DOL products, the ampicillin was quickly degraded such that the cells were rescued. In comparison, cells were completely eliminated due to the presence of antibiotics (Fig. 5c). The same DOL product was also able to convert the highly toxic paraoxon (PAR) to less harmful paranitrophenol (PNP), as reflected by the increment in the characteristics absorbance of PNP during the reaction (Fig. 5d and Supplementary Fig. 11b).

Members inside the MSBC can communicate. To this notion, we engineered a pair of sender and receiver MSBs. The *E. coli* inside the sender MSB were inducible to express 3-oxo-$C_{12}$-HSL (3OC12), which could diffuse across the bacteria membranes and polymeric capsules. 3OC12 was then sensed by the corresponding allosteric transcription factors (VP16-LasR) of the *S. cerevisiae* inside the receiver MSB[36,37], triggering the expression of the YFP (Fig. 6a). Again, MSBC, instead of the normal well-mixed consortia, could tightly regulate the composition of the sender and receiver cells (Fig. 6b and Supplementary Figs. 12 and 13). Since the concentration of the signaling molecules is determined by the percentage of the senders, we could readily modulate the strength of the communication inside MSBC. By manipulating the seeding ratio of sender and receiver MSBs, we have managed to gradually activate the YFP expression inside the consortia (Fig. 6c, d). This trend was resembled with in vitro titration experiment, emphasizing the controllability over the signal strength in communication by MSBC platform (Supplementary Fig. 14). MSBC platform could also modulate the signal range during the communication. We fabricated microdevices with two chambers containing the MSB$_{sender}$ and MSB$_{receiver}$, connected by channels at different lengths. Results showed that regardless of the distances (from 0 to 30 mm), cells of MSB$_{receiver}$ were all activated by MSB$_{sender}$ (Supplementary Fig. 15). These results reinforced the capabilities of MSBC platform in tuning both the signal strength and range of the communication inside the communities.

**MSBC strategy enables the assembly of phototrophic microbial communities.** Microbial communities comprised of phototrophs exhibit symbiosis in nature since the photoautotrophic members, such as cyanobacteria can convert carbon dioxide ($CO_2$) into organic carbon to maintain the growth of the heterotrophic members. Therefore, engineering the synthetic phototrophic communities holds great promise for bioenergy, such as the green manufacturing of the biomass and biofuels. Here, we further assembled a phototrophic MSBC such that the MSB$_{photoautotroph}$ could produce sucrose from $CO_2$ and supported the growth of the MSB$_{heterotroph}$ (Fig. 7a). To implement the design, we first generated the MSB(*S. elongatus*) by encapsulating *S. elongatus* PCC 7942 with alginate, and cultured these MSB under an illuminating incubator. The color of MSB(*S. elongatus*) turned green gradually with time, suggesting the growth of the cyanobacteria inside the polymeric capsules (Fig. 7b). We programmed the *S. elongatus* PCC 7942 to express the sucrose permease such that the *S. elongatus* could secrete sucrose under the osmatic stress. The MSB(*S. elongatus*) stressed by the NaCl treatment produced the sucrose, as shown in Fig. 7c. It is noted that the yield of sucrose for MSB(*S. elongatus*) is ~73% higher than that of the free cells (Supplementary Fig. 16). This is possibly caused by an enhanced metabolic activity of *S. elongatus* by immobilization[38–40]. To ensure

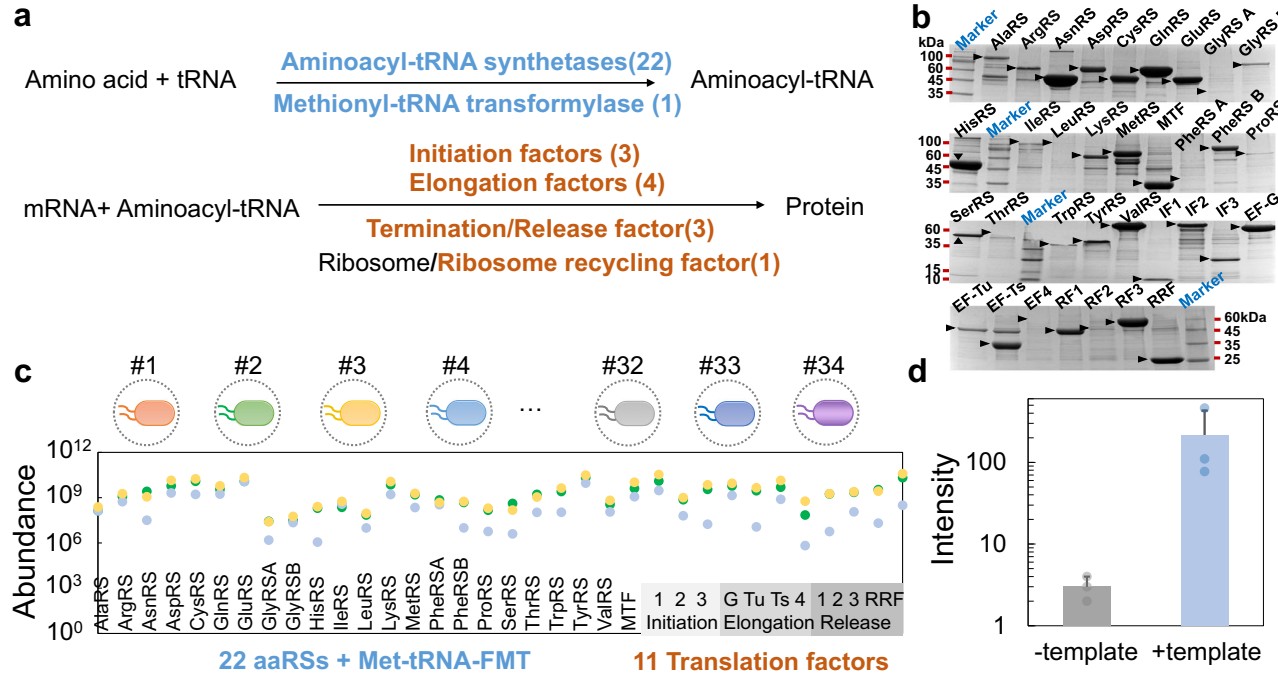

**Fig. 3 34-strain microbial swarmbot consortium enables the assembly of core machinery of PURE. a** The core machinery of protein synthesis in *E. coli* requires 34 enzymes. In the first step of protein synthesis, 22 aminoacyl-tRNA synthetases (Phe-tRNA synthetase and Gly-tRNA synthetase have two subunits for each) catalyze the aminoacylation reaction by covalently linking an amino acid to its cognate tRNA. Methionyl-tRNA transformylase attaches a formyl group to the free amino group of methionyl-tRNA. The translation factors responsible for initiation, elongation and termination are three initiation factors (IF1, IF2, and IF3), four elongation factors (EF-G, EF-Tu, EF-Ts, and EF-4), three release factors (RF1, RF2, and RF3) and ribosome recycling factor (RRF). **b** Individually cultured 34 different MSBs produced each of the 34 enzymes. A circuit (ePop) programmed the density-dependent autonomous lysis of the *E. coli*. The cells were trapped inside the polymeric scaffold, while the proteins were transported to the exterior of the MSB. The supernatant was collected and purified for examination. SDS-PAGE showed the expression of 34 proteins by individually cultured MSBs. Experiments were repeated independently more than three times with similar results. **c** Mass spectrometry results suggested the presence of all 34 elements, as purified from the supernatant of the microbial swarmbot consortium (MSBC). We co-cultured the MSBC consisting of 34 MSBs, collected the supernatant and purified it with His-tagged resins. The purified elements were subject to the mass spectrometry quantification. The *y* axis is in arbitrary units. **d** Reconstitution of the reaction using the MSBC produced 34 enzymes led to the RFP expression. Addition of the template DNA (+template) drove the expression of RFP in the reaction. When the reaction is not supplemented with template DNA (−template), there was no signal detected. The *y* axis is in log(arbitrary unit). Error bars = Standard Deviation (*n* = 3 biologically independent samples). Source data are provided as a Source Data file.

that the *E. coli* could utilize the sucrose, we programmed the cells with a circuit containing the essential genes (*cscB*, *cscK*, and *cscA*) for sucrose metabolism. Uptake of sucrose across the cell membrane was assisted by the sucrose permease (encoded by *cscB*). The sucrose was then split into glucose and fructose by the invertase (encoded by *cscA*), while the fructose was further phosphorylated by fructokinase (encoded by *cscK*). Indeed, *E. coli* (MG1655) carrying the circuit grew by digesting the sucrose (Fig. 7d and Supplementary Fig. 17). We then paired the MSB(*S. elongatus*) and MSB(*E. coli*) to form the MSBC. In minimal media devoid of any organic carbon source, the phototrophic MSB(*S. elongatus*) successfully sustained the heterotrophic MSB(*E. coli*) at a high MSB(*S. elongatus*) seeding density, as evidenced by the presence of multiple *E. coli* colonies inside MSB(*E. coli*) (Fig. 7e, f and Supplementary Figs. 17 and 18).

## Discussion

In this study, we developed an engineered spatial segregation strategy to construct microbial consortia in a flexible and precise manner. Previous studies have used membrane-embedded or specifically designed (e.g., fabrication of hundred nanometers deep nanoslits which allow for the diffusion of nutrients, metabolites and signaling molecules while being too shallow for bacteria to pass through) microfluidic chips to investigate the spatial organization of the microbial consortia[7,12,41–44]. While the design of these chips is intuitive, nevertheless, the strategies are generally not scalable for

synthetic biology implementations due to the limited volume (usually below 1 mL)[45,46], the complicated fabrication and operation processes (especially for the membrane-embedded chips)[41,47,48], and the lack of the modularity (have to frequently modify the design based on the type of the consortia and the aims)[12,42,43]. By design, MSBC is scalable and modular. The MSBC platform can be easily scaled up by adjusting the number of different MSBs and the culture volume, depending on the specific applications (Figs. 3–6). The consortia can be built with various MSBs containing different strains or species based on design. The host microbes and the encapsulating material can be separately engineered or optimized and then integrated. These properties make our platform highly versatile and flexible: for example, we demonstrated the assembly of multiple two-species or three-species consortia without extensive optimization in each case (Figs. 4–6), suggesting that the desired consortia can be customized by simply interchanging the MSB members. This plug-and-play concept will be particularly effective to build a wide range of consortia.

By far, relatively few studies have been devoted to developing systems that could regulate the compositions of subpopulations within consortia, despite that the ratio in the subpopulation is often crucial in the coordinated tasks. Recently, there are several works re-engineered the native cell–cell signaling systems to enable the autonomous coordination of subpopulation densities. These systems often include a sophistically designed cell-based signal translator and growth-controller modules, both with

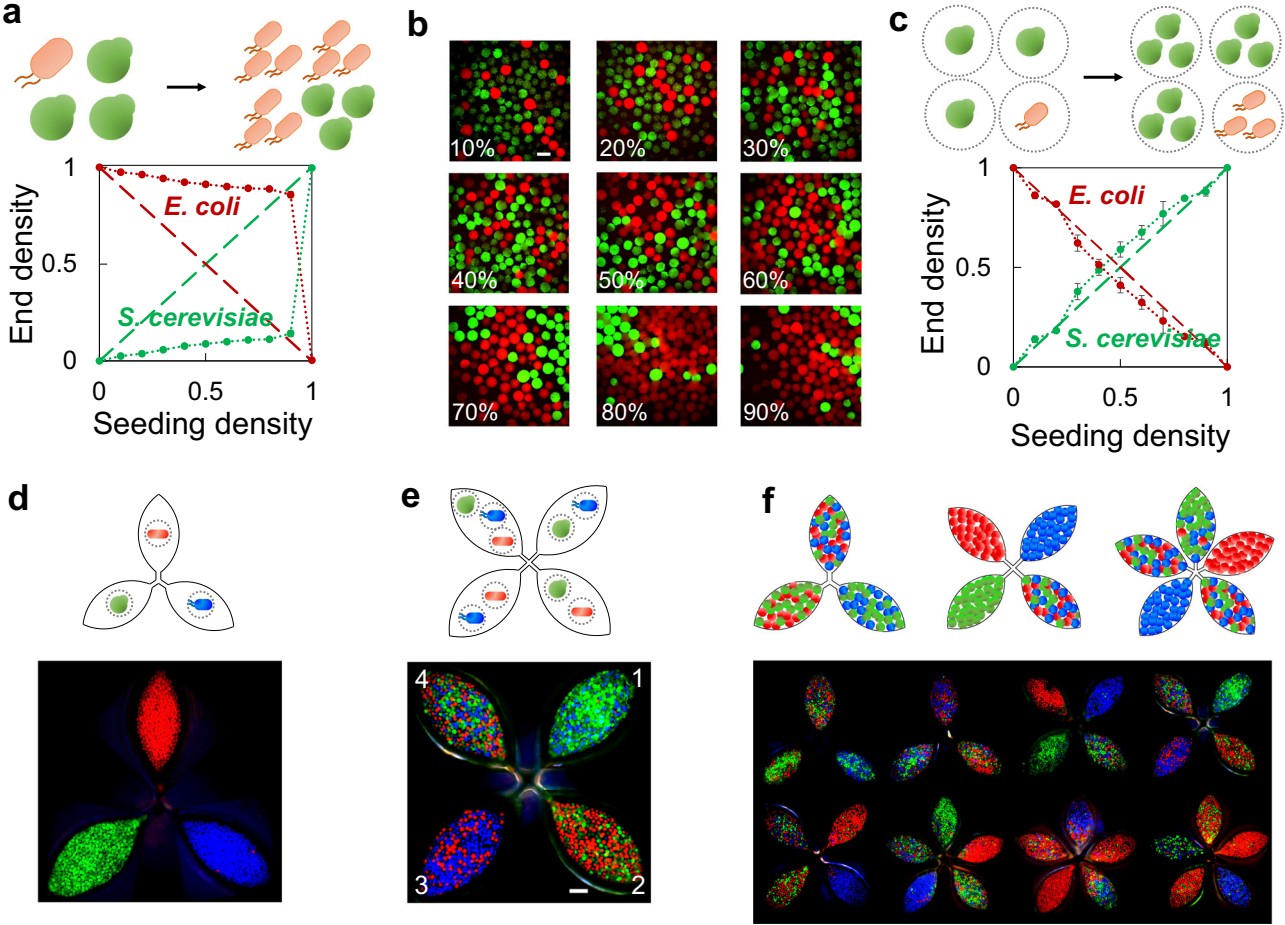

**Fig. 4 MSBC platform built multispecies consortia with flexibility and precision. a** Faster-growing *E. coli* overtook the *S. cerevisiae* in a well-mixed consortium. Cell mixture (*S. cerevisiae* (BY4742 expressing GFP) and *E. coli* (MG1655 expressing mCherry) at different seeding ratios) were inoculated into the medium, cultured at 30 °C for 24 h and quantified by flow cytometry. The dotted and dashed lines showed the actual and ideal composition, respectively. The *x* axis indicated the seeding density of *S. cerevisiae* ($\frac{N_{S.cerevisiae}}{N_{S.cerevisiae}+N_{E.coli}}$). In all conditions, the faster-growing *E. coli* became dominant. **b** The growth of *E. coli* and *S. cerevisiae* in MSBC was constrained inside their own niche with a defined boundary. We mixed the MSB(*S. cerevisiae*) and MSB(*E. coli*) at different seeding ratios (~2000 MSBs in total), and cultured the MSBC at 30 °C for 24 h. In all conditions, two species grew inside their own MSBs without interference and were able to maintain a desired population ratio consequently. Numbers indicated the seeding density of MSB(*E. coli*) ($\frac{MSB_{E.coli}}{MSB_{S.cerevisiae}+MSB_{E.coli}}$). Scale bar = 400 μm. **c** MSBC enabled precise control in the composition. The dotted and dashed lines showed the actual and ideal composition, respectively (see "Methods"). Quantification results reflected that MSBC strategy could precisely modulate the composition of the consortia. The *x* axis indicated the seeding density of MSB(*S. cerevisiae*). Error bars = standard deviation (*n* = 3 biologically independent samples) in **a**, **c**. **d** The growth of various species were supported in MSB. We generated MSB (*E. coli*) (expressing BFP), MSB(*S. cerevisiae*)(expressing GFP) and MSB(*C. glutamicum*) (expressing RFP), respectively. All three species grew inside MSBs showing the desired fluorescence. **e** MSBC platform assembled multispecies consortia. We randomly assembled three two-species MSBC (chamber 1–3, seeding ratio = 1:1), and a three-species MSBC (chamber 4, seeding ratio = 1:1:1). In all chambers, multiple species co-existed and the consortia composition was precisely modulated by the seeding ratio. Scale bar = 1 mm. **f** Assembly of consortia can be customized by MSBC strategy. According to the designs (schematic on top), we were able to construct various MSBC by simply replacing the MSB of different species. Source data are provided as a Source Data file.

relatively large genetic parts[49,50]. This poses two issues. First, the genetic payload of the consortia significantly increases, consequently compressing the resources for the rest missions (usually the missions that need composition control). Also, the limited orthogonal communication tools and efficient control methods weigh down the diversity of the consortia. In this respect, MSBC method does not consume the genetic resources of the microbes or suffer from the constraint in the toolboxes of communication and control. The physical partition strategy offers a generalizable platform for consortia assembly and composition control.

Polymeric capsules provide physical support to the living cells, restrain the free diffusion of cells and fence the subpopulations. The escape rate of MSB(*E. coli*) and MSB(*S. cerevisiae*) was ~ 0.2% and 0.004%, respectively. The modification on the surface

of MSB could decrease the escape rate of MSB(*E. coli*) to zero after 24 h of culture (Supplementary Fig. 19a). The system can consistently operate for 8 days (Supplementary Fig. 19b). Its stability offers a side benefit: these MSBs can be prepared in advance and stored in long term with a negligible decrease in performance (Supplementary Fig. 20). The study presented here offers a fresh method for engineering and controlling microbial consortia, which would bring insights into areas including biomanufacturing, bioremediation, and bioenergy.

## Methods

**Bacterial strains.** *Escherichia coli* strain BL21(DE3) carrying the autolysis circuit was used for protein expression (Figs. 2c, 3, and 5). *E. coli* strain MG1655 constitutively expressing mCherry was used to generate (Figs. 2a and 4a–c). *E. coli*

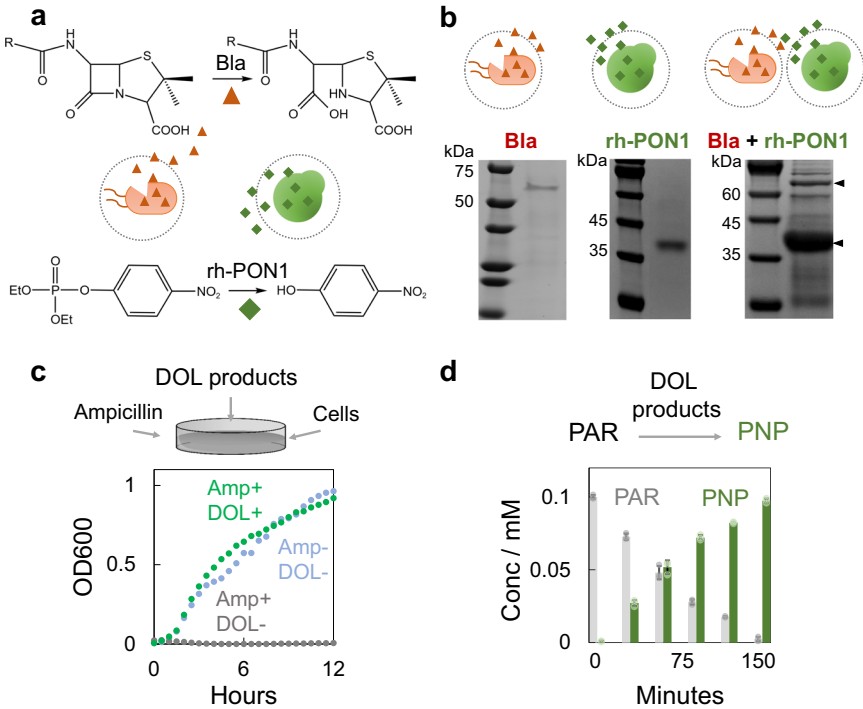

**Fig. 5 MSBC platform enabled the controllable division of labor across the species. a** Schematic shows that division of labor (DOL) product of the engineered MSBC can degrade both antibiotics and organophosphates. We engineered the MSB(*E. coli*) that can release β-lactamase (Bla) through autonomous cell lysis, and the MSB(*P. pastoris*) that can secret recombinant human paraoxonase (rh-PON1). The DOL product of MSBC contained both β-lactamase and rh-PON1, which could collectively degrade a broad class of β-lactams antibiotics by breaking the β-lactam ring and detoxify the organophosphate insecticides such as paraoxon by hydrolysis reaction. **b** MSBC produced both enzymes simultaneously. We cultured the MSB(*E. coli*), MSB(*P. pastoris*) or MSBC (containing both MSBs) in the nutrients for 48 h, collected the supernatants and purified for SDS-PAGE analysis. Results showed the presence of β-lactamase (Bla) or recombinant human paraoxonase (rh-PON1) manufactured by the individually cultured MSB(*E. coli*) or MSB(*P. pastoris*) (left and middle gel, 10 mL cell culture). Culturing the MSBC led to the simultaneous generation of two enzymes (right gel, 50 mL cell culture). Experiments were repeated independently more than three times with similar results. **c** DOL product of MSBC degraded antibiotics and rescued the cells. In all, 5 μL purified enzymes mixture (from the supernatant of MSBC) was added into the 200 μL LB medium with 300 μg/mL ampicillin. Cells that are sensitive to the ampicillin (DH5α) were inoculated into the medium immediately (1:200 ratio) and the absorbance at 600 nm was monitored by a platereader. The growth of the rescued cells (green curve) was hardly influenced and comparable with the control without adding antibiotics (blue curve). In contrast, the growth of the un-rescued cells was completely eliminated due to the presence of antibiotics (gray curve). **d** DOL product of MSBC converted the paraoxon (PAR) to paranitrophenol (PNP). The highly toxic PAR was hydrolyzed to PNP by the DOL product gradually. The concentration of PNP was quantified based on its absorbance at 405 nm. The consumption of PAR was then calculated based on the generation of PNP. Error bars = standard deviation (*n* = 3 biologically independent samples). Source data are provided as a Source Data file.

strain BL21(DE3) transformed with an inducible TagBFP plasmid was used to generate (Fig. 4d–f). *Pichia pastoris* X-33 was used for secretory protein expression (Figs. 2d and 5b–d). *Saccharomyces cerevisiae* BY4742 constitutively expressing eGFP was used to generate (Figs. 2b and 4). *Corynebacterium glutamicum* ATCC 13032 carrying a plasmid (XK99E) that programmed the constitutive mCherry expression was used to generate (Fig. 4d–f). *E. coli* strain MG1655 transformed with a plasmid CHP1 (pBAD-LasI) that produces the 3-oxo-C12-HSL (3OC12) under induction was used as the sender strain, and *S. cerevisiae* CYE72(BY4741 xylR natMX tetR lacI sensor strain) whose genome was integrated with the $P_{lasO}$-YFP and the $P_{tet}$-VP16-LasR operator was used as the receiver strain (Fig. 6)[51]. *S. cerevisiae* yCAN14 that can use olivetolic acid to produce cannabigerolic acid was used to generate (Fig. 2e)[29]. *Synechococcus elongatus* PCC 7942 expressing sucrose permease (coded by *cscB* gene) under the induction and *E. coli* MG1655 transformed with a circuit that constitutively expressing the sucrose metabolism genes (*cscB* (ECW_m2594), *cscK* (ECW_m2595) and *cscA* (ECW_m2596)) were used to generate (Fig. 7)[52].

**Circuit and plasmids**. ePop (ColE1 origin) was published previously[53]. Briefly, it was constructed using the *luxbox* region (140 bp upstream of *luxI* in *V. fischeri*) from p*lux*GFPuv and *E* gene coding sequence from φX174 (NEB). Each region was PCR-amplified and then joined together in an overlap PCR reaction. The "*lux* box-*E* gene" fragment was inserted into the AatII site of host vector pLuxRI2.

Elastin-like polypeptide (ELP)-fused GFP-His was constructed using the plasmids from Prof Chilkoti's lab. The fragments were cloned into the vector with the T7 promoter and p15A origin.

The 34 translation machine genes were synthesized by Genewiz and cloned into the vector with a T7 promoter and p15A origin with a 6×His-tag located at either the N or C terminal[15].

Bla was constructed using the synthesized fragment (His-ELPs-BlaM) from Genewiz. The fragment was cloned into the vector with the T5 promoter and p15A origin.

Recombinant human growth hormone (rh-GH) and recombinant human paraoxonase 1 (rh-PON1) were codon-optimized for expression in *P. pastoris* and synthesized by Genewiz[35]. The constructs were assembled into plasmid pPICZα A (purchased from Thermo Fisher (V19520)), and integrated into the genome of *P. pastoris* according to the product manual.

pBAD-LasI was constructed using a codon-optimized LasI (synthesized by BGI). This plasmid was transformed into *E. coli* MG1655 (sender cells).

$P_{tet}$-VP16-LasR and $P_{lasO}$-YFP were codon-optimized for expression in *S. cerevisiae* and synthesized by BGI. These operons were integrated into the genome of *S. cerevisiae* CYE72 (receiver cells).

The *cscB* gene from *E. coli* W was overexpressed in *S. elongatus* PCC 7942 under the theophylline-inducible promoter $P_{theo}$ for efficient sucrose secretion. The expression cassette was integrated into the *S. elongatus* PCC 7942 genome using integration plasmid PSI[54]. We have summarized the protein sequences of the essential constructs in Supplementary Table 1.

**Luria-Bertani medium**. A total of 25 g Luria-Bertani (LB) broth powder (Aladdin, Shanghai, China) was added into 1 L deionized $H_2O$. After autoclaving for 45 min, the LB medium was stored at room temperature.

**M9 medium**. A total of 6.65 g of M9CA broth medium (Sangon Bioiotech) was added to 500 mL deionized $H_2O$ (adjusting pH to 7.4) for autoclave. 2 mM $MgSO_4$,

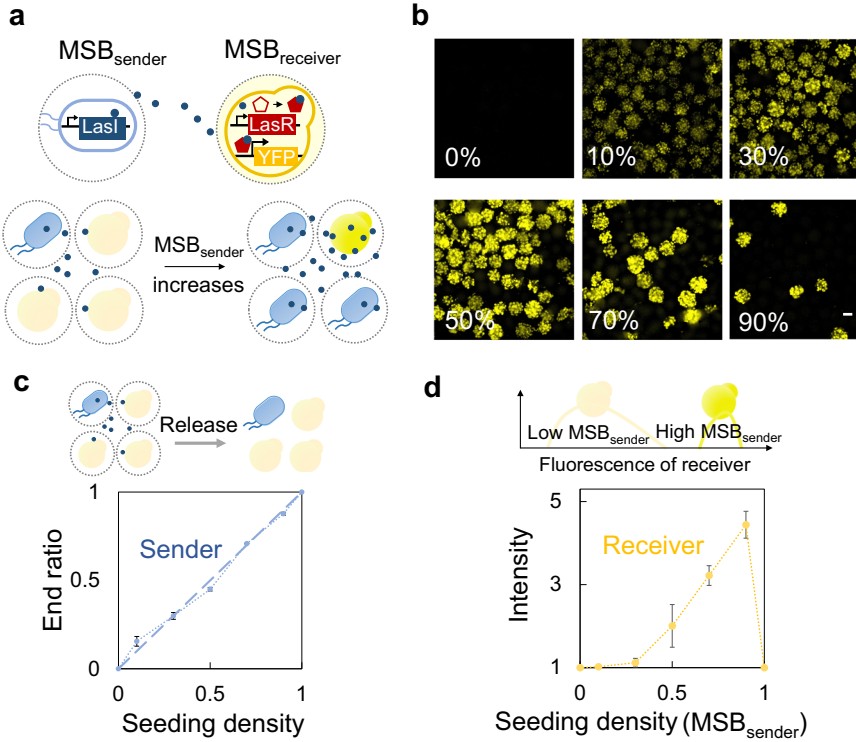

**Fig. 6 MSBC platform enabled communication across the species. a** Schematic shows that MSBC platform could modulate the communication across the species. We programmed the *E. coli* to express 3-oxo-$C_{12}$-HSL (3OC12) under induction and generated the MSB$_{sender}$. We programed the *S. cerevisiae* with a circuit bearing the corresponding inducible allosteric transcription factors (VP16-LasR), and generated the MSB$_{receiver}$. The signals (3OC12) of MSB$_{sender}$ diffused across the cell membranes and polymeric capsules, and were sensed by the MSB$_{receiver}$ to drive the expression of the YFP. MSBC could tightly module the signal intensity. A higher MSB$_{sender}$ seeding density led to an increased concentration of signaling molecules, which triggered a stronger expression of YFP of MSB$_{receiver}$. **b** Increasing the seeding density of MSB$_{sender}$ gradually lighted up the MSB$_{receiver}$. MSB$_{sender}$ and MSB$_{receiver}$ were co-cultured with different seeding ratios (~2000 MSBs in total) into the medium (1 mL LB+SC). The number on the images indicated the seeding density of MSB$_{sender}$ ($\frac{MSB_{sender}}{MSB_{sender}+MSB_{receiver}}$). The resultant MSBC were all cultured at 30 °C for 24 h. There were hardly any YFP signals in the absence of MSB$_{sender}$. Increasing the MSB$_{sender}$ density gradually lighted up the MSB$_{receiver}$ because of the increased concentration of signaling molecules (3OC12). Scale bar = 200 μm. **c** Quantification results reflected that MSBC strategy precisely modulated the composition of consortia. The cells of MSBC were released from the polymeric capsules and counted by the flow cytometry measurement. The *x* axis indicated the seeding density of MSB$_{sender}$. Error bars = standard deviation (*n* = 3 biologically independent samples). **d** Quantification results reflected that MSBC strategy tightly modulated the signal intensity. Cells were released from the MSBC. YFP intensity was quantified through the flow cytometry (the median of the population) and normalized by the value when seeding density of MSB$_{sender}$ equals to 0. The *x* axis indicated the seeding density of MSB$_{sender}$. Error bars = standard deviation (*n* = 3 biologically independent samples). Source data are provided as a Source Data file.

0.1 mM $CaCl_2$, and 0.2% glucose were further added into the medium. The medium was stored at 4 °C.

**Synthetic complete medium.** A total of 1.7 g yeast basic nitrogen source (YNB without amino acids and ammonium sulfate, BD), 5 g ammonium sulfate (VETEC™, Merck), 20 g D-glucose, 0.1 g L-arginine, 0.1 g L-cysteine, 0.1 g L-lysine, 0.1 g L-threonine, 0.05 g L-aspartic acid, 0.05 g L-Isoleucine, 0.05 g L-phenylalanine, 0.05 g L-proline, 0.05 g L-serine, 0.05 g L-tyrosine, 0.05 g L-valine, 0.05 g L-methionine, 0.1 g L-tryptophan, 0.05 g L-histidine, 0.1 g L-uracil, 0.1 g L-leucine, 0.1 g L-adenine were added to 1 L deionized $H_2O$. All amino acids were purchased from Sigma-Aldrich. After autoclaving for 45 min, the SC medium was stored at room temperature. In all, 2% (m/v) glucose was supplemented to the medium before use.

**YPD medium.** A total of 10 g of yeast extract (OXOID, LP0021) and 20 g peptone (Bacto™, BD) were added to 1 L deionized $H_2O$. After autoclaving for 45 min, YPD medium was stored at room temperature. In total, 2% (m/v) glucose was supplemented to the medium before usage.

**The BM(M/G)Y medium.** A total of 10 g yeast extract, 20 g peptone, 3.4 g yeast basic nitrogen source (YNB) and 10 g ammonium sulfate were added to 900 mL deionized $H_2O$ and autoclaved for 45 min, and then mixed with 100 mL of separately sterilized 1 M potassium phosphate buffer (pH 6.0, Sangon Biotech, Shanghai) with $4 \times 10^{-5}$ % (m/v) biotin (Sangon Biotech, Shanghai) to prepare the BMY medium. In all, 1% (v/v) glycerin (Lingfeng, Shanghai) or 1% (v/v) methanol (Sangon Biotech, Shanghai) was added to the BMY medium to prepare BMGY or BMMY medium.

**The BG-11 medium.** A total of 0.425 g of BG-11 medium powder (Hopebio) was added to 250 mL deionized $H_2O$ for autoclave. The medium was sealed and stored at room temperature.

**The CoBG-11 medium.** Co-culture medium (named CoBG-11) was designed based on BG-11 medium and optimized for *E. coli* growth by supplementing with 106 mM NaCl, 4 mM $NH_4Cl$ and 25 mM N-(2-hydroxyethyl)-piperazine-N'-2-hydroxypropanesulfonic acid (HEPPSO). The pH value of CoBG-11 was adjusted to 7.5 by KOH. NaCl and $NH_4Cl$ were added to maintain the viability of *E. coli*. NaCl (150 mM) was also used as a stress inducer for sucrose accumulation in *S. elongatus*.

All medium used was supplemented with appropriate antibiotics (50 μg/mL kanamycin, 75 μg/mL ampicillin, 100 μg/mL chloramphenicol, 50 μg/mL zeocin, and 50 μg/mL spectinomycin) when applicable. IPTG at 0.1 or 1 mM, theophylline at 2 mM, Anhydrotetracyclin (ATc) at 100 ng/mL or arabinose at 5 mM was used to induce gene expression when applicable. In all, 2% Agar (Bacto™, BD) was dissolved in the liquid medium to prepare the related agar plate.

**Overnight liquid culture.** *E. coli* carrying the ePop circuit were streaked onto LB agar plate supplemented with 2% (w/v) glucose and incubated at 37 °C for 16 h. Then, a single colony was picked and inoculated in 3 mL LB medium. The agar and the liquid medium were supplemented with appropriate antibiotics when applicable. For bacteria carrying ePop circuit, 2% (w/v) glucose was supplemented in the overnight culture.

*S. cerevisiae* were streaked onto YPD agar plate and incubated at 30 °C for 48 h. Then, a single colony was picked and inoculated in 3 mL SC medium for 24 h.

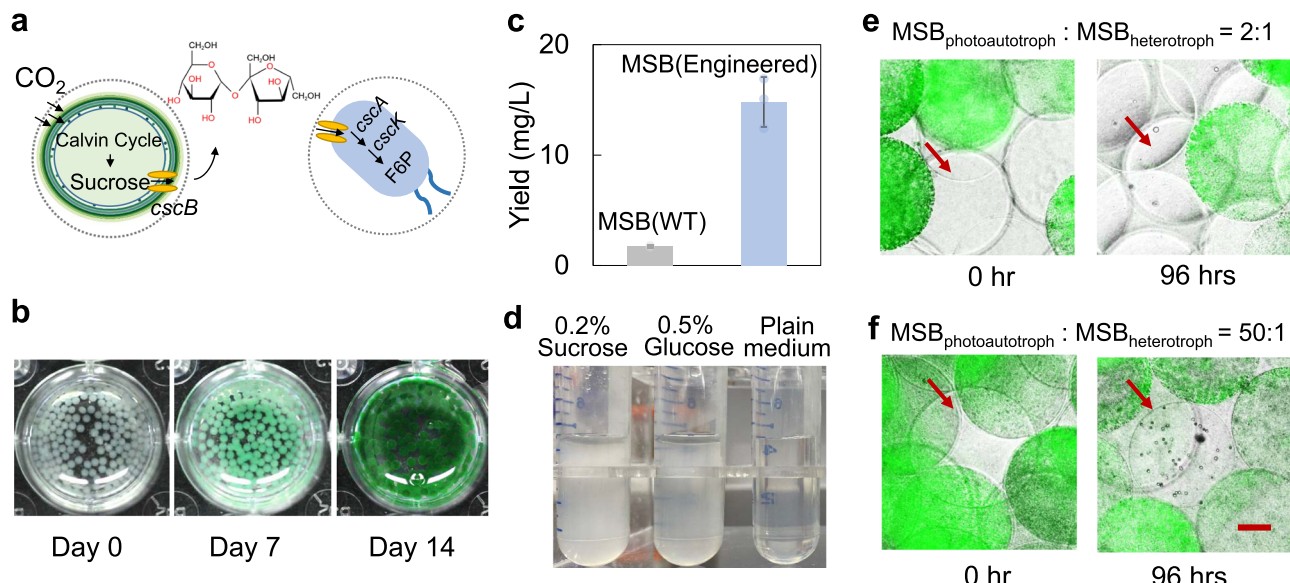

**Fig. 7 MSBC enabled phototrophic community assembly. a** Schematic shows that the MSBC can be phototrophic. MSB$_{phototroph}$ containing the sucrose-secreting *S. elongatus* convert the carbon dioxide into organic carbon and sustain the growth of MSB$_{heterotroph}$. **b** Generation of MSB(*S. elongatus*). We encapsulated the *S. elongatus* with polymeric capsules made of alginate. The growth of the cyanobacteria turned the MSB(*S. elongatus*) into an enhanced green color. **c** MSB(*S. elongatus*) produced sucrose under the osmotic stress induced by sodium chloride. The MSB(*S. elongatus*) were cultured in BG-11 medium and stressed by supplementing 150 mM NaCl. The yield of sucrose was measured by a sucrose quantification kit (see "Methods"). Error bars = standard deviation (*n* = 3 biologically independent samples). **d** Engineered *E. coli* grew by using sucrose as a carbon source. *E. coli* were engineered to carry the essential sucrose metabolism genes. The resultant strain utilized sucrose as a carbon source for growth. **e** Assembly of phototrophic MSBC with a low seeding density of MSB$_{phototroph}$. When the seeding ratio of MSB$_{phototroph}$ and MSB$_{heterotroph}$ equals 2:1, the low density of MSB$_{phototroph}$ only supported the limited growth of the MSB$_{heterotroph}$. **f** Assembly of phototrophic MSBC with a high seeding density of MSB$_{phototroph}$. When seeding ratio of MSB$_{phototroph}$ and MSB$_{heterotroph}$ equals 50:1, the high density of MSB$_{phototroph}$ better supported the growth of the MSB$_{heterotroph}$, as shown by the increased density of the *E. coli* colonies inside the MSB. Scale bar = 150 μm. Source data are provided as a Source Data file.

*C. glutamicum* were streaked onto LB agar plate and incubated at 30 °C for 16 h. Then, a single colony was picked and inoculated in 3 mL LB medium for 24 h.

*P. pastoris* were streaked onto YPD agar plate and incubated at 30 °C for 48 h. Then, a single colony was picked and inoculated in 3 mL BMGY medium for 24 h.

*S. elongatus* were cultured in BG-11 medium under a light intensity of ~100 μmol photons m$^{-2}$ s$^{-1}$ at 30 °C (shaking speed = 130 rpm) in an illuminating incubator (GXM-508-4, Ningbo JIANGNAN Instrument). To maintain the stable phenotype of sucrose secretion, appropriate antibiotics were added when necessary.

**Production of chitosan microcapsules containing microbes.** Chitosan or alginate microcapsules containing microbes were generated by electrospray. A homogeneous mixture of 2% (w/v in 1% acetic acid solution) chitosan solution or 2% (w/v in D.I. water) sodium alginate solution (both from Sigma-Aldrich) and microbes were transferred into a syringe with an attached blunt tip and sprayed at 5 kV. The generated chitosan or alginate microsphere were cross-linked in the 5% w/v tripolyphosphate (Sigma-Aldrich) or 1.5% (w/v) calcium chloride for 30 min.

**Generation of MSB(*E. coli*) and MSB(*S. cerevisiae*).** *E. coli* strains were pre-grown in LB medium. 1 mL of overnight culture were centrifuged, resuspended in 100 μL LB medium, and encapsulated into 1 mL chitosan solution to generate MSB(*E. coli*). ~30,000 MSB(*E. coli*) were inoculated into 10 mL LB medium at 37 °C. *S. cerevisiae* BY4742 were pre-grown in SC medium for 24 h. 1 mL of overnight culture were centrifuged and resuspended in 100 μL SC medium, and encapsulated into 1 mL chitosan solution to generate MSB(*S. cerevisiae*). ~30,000 MSB(*S. cerevisiae*) were inoculated into 10 mL SC medium at 30 °C.

**Protein production by MSB(*E. coli*).** *E. coli* BL21(DE3) carrying both the autolysis circuit and protein expression circuit were pre-grown in LB medium with 2% glucose for overnight. 1 mL of overnight culture were centrifuged, resuspended in 100 μL LB medium and encapsulated into 1 mL chitosan solution to generate MSB(*E. coli*). ~30,000 MSB(*E. coli*) were inoculated into 10 mL LB medium at 37 °C. The supernatant of the MSB culture was collected, purified and analyzed according to the experimental requirement.

**Protein production by MSB(*P. pastoris*).** *P. pastoris* X-33-rh-GH/X-33-rh-PON1 were pre-grown in BMGY medium for 24 h. Cells of the overnight culture were collected by centrifugation and starved in BMY medium for 1 h. In all, 1 mL of overnight culture were centrifuged, resuspended in 100 μL BMY medium and

encapsulated into 2 mL chitosan solution to generate MSB(*P. pastoris*). In all, ~30,000 MSB(*P. pastoris*) were inoculated into 10 mL BMMY medium and cultured in a 50 mL Erlenmeyer flask at 30 °C for 120 h (shaking speed = 200 rpm). 1% (v/v) anhydrous methanol was added every 24 h.

**Protein purification.** For His-tag affinity purification, we used 75 μL cOmplete™ His-Tag purification resin (Sigma-Aldrich) to bind 4 mL supernatant and eluted the resin with 75 μL elution buffer. For all protein samples, SDS-PAGE with Coomassie Blue stain was used to verify the proteins.

**Evaluating the protein size that could be exported from capsules.** A high molecular weight (~150,000 g/mol) rhodamine-labeled dextran (Sigma-Aldrich) was used as a model protein. We mixed 30 μL of dextran–rhodamine (20 mg/mL) with 300 μL chitosan and generated the capsules. The chitosan capsules were first washed by PBS for three times and then cultured in 3 mL M9 medium. As control, same amount of plain capsules were cultured in 3 mL M9 media. In total, 100 μL of the culturing medium was taken at different time points and the fluorescence signal from the medium was measured by a platereader (Infinite 200pro, TECAN). Experiments were done in triplicate and measurements were background corrected. Unless mentioned otherwise, plots in main figures and supplementary figures were generated by Excel (2013).

**Scanning electron microscope observation.** Chitosan capsules were first frozen in liquid nitrogen and then lyophilized overnight (Christ Alpha1-2LDplus). The capsules were dissected and then observed under the scanning electron microscope (Phenom Pharos Desktop SEM).

**CBGA production by MSB(*S. cerevisiae*) and free cells.** *S. cerevisiae* yCAN14 was pre-grown in YPD medium overnight. In total, 1 mL of overnight culture were centrifuged, resuspended in 100 μL SC medium, and encapsulated into 1 mL chitosan solution to generate MSB(*S. cerevisiae*). ~30,000 MSB(*S. cerevisiae*) or 100 μL yCAN14 overnight culture were inoculated into 10 mL SC medium and cultured at 30 °C for 120 h (shaking speed = 200 rpm). Galactose (2% (w/v)) was supplemented every 24 h and 1 mM olivetolic acid was supplemented to the MSB or the free-cells culture after 24 h.

**CBGA extraction**. Overall, ~12,000 MSB(*S. cerevisiae*) were treated with 2 U/OD zymolyase (Zymo Research) by shaking at 800 rpm for 2 h (30 °C), followed by ethyl acetate/formic acid (0.05% (v/v)) extraction in a 2:1 volume ratio with bead-beating (30 s⁻¹, 3 min). Organic and inorganic layers were separated by centrifugation at 12,000 × *g* for 1 min. Samples were extracted three times. The combined organic layers were evaporated in a concentrator plus at 45 °C (V-AL, Eppendorf) and the remainders were resuspended in 200 μL acetonitrile/H₂O/formic acid (80%/20%/0.05% (v/v/v)). Finally, samples were filtered with Ultrafree-MC columns (0.22-μm pore size, polyvinylidene difluoride (PVDF) membrane).

**HPLC and LC–MS analysis**. Products were analyzed using both high-performance liquid chromatography with UV detection (HPLC–UV, Agilent 1260 Infinity II) and LC/MS (Agilent 6470B LC/TQ, Agilent Technologies) equipped with a reverse phase C18 column (Symmetry C18 3.5 μm, 100 × 2.1 mm Column, Waters; Kinetex 2.6 μm F5 100 Å 100 × 2.1 mm, Phenomenex, respectively). The mobile phase was composed of 0.05% (v/v) formic acid in water (solvent A) and 0.05% (v/v) formic acid in acetonitrile (solvent B). Cannabinoids (CBGA) were separated in HPLC via gradient elution as follows: linearly increased from 30% B to 80% B in 13 min, held at 80% B for 5 min, increased from 80% B to 97% B in 0 min, held at 97% B for 2 min, decreased from 97% B to 30% B in 0.1 min, and held at 30% B for 2.9 min. The flow rate was held at 0.3 mL min⁻¹. The total liquid chromatography run time was 23.0 min. And the analysis method of LC/MS is the same as described previously[29].

The samples were separated in LC/MS via gradient elution as follows: linearly increased from 20% B to 40% B in 5 min, then increased to 80% B in 13 min, increased from 80% B to 97% B in 2 min, held at 97% B for 1 min, decreased from 97% B to 20% B in 0.1 min, and held at 20% B for 2.4 min. The flow rate was held at 0.3 ml min⁻¹ and the total liquid chromatography run time was 18.5 min.

Sample injection volumes of 10 μL and 1 μL were used for HPLC and LC–MS, respectively. The sample tray and column compartment were set to 15 °C and 40 °C, respectively. For HPLC–UV, CBGA was detected by diode array detection at 210/270/325 nm. For LC/MS, electrospray ionization was conducted via the Agilent Jet Stream thermal gradient focusing technology, in which the sheath gas flow rate and temperature were set to 11 L min⁻¹ and 250 °C, respectively. Drying and nebulizing gases were set to 5 L min⁻¹ and 45 psi, respectively, and a drying-gas temperature of 300 °C was used throughout. All other conditions were the same, as described previously[29]. Data files were processed by Agilent MassHunter Qualitative Analysis software.

**Enzymes manufactured by MSBC**. A total of 34 different *E. coli* BL21(DE3) carrying both the autolysis circuit and protein expression circuit were individually pre-grown in LB medium with 2% glucose overnight. In all, 4 mL of overnight culture were centrifuged, resuspended in 400 μL LB medium, and encapsulated into 4 mL chitosan solution to generate MSB(*E. coli*). The seeding density of MSBs expressing GlyRS A and B, pheRS A and B, EF-Tu, IF1, ProRS, and LeuRS were 8% (v/v), while the seeding density of the other 26 MSBs was 1.4% (v/v). The MSBC were cultured in fresh M9 medium and induced with 0.1 mM IPTG for 24 h at 37 °C by shaking. The supernatant was harvested and filtrated through 0.22-μm filter, mixed with Ni Sepharose 6 fast flow (GE Healthcare), and incubated at 4 °C for 1 h. Unbound proteins were washed out by buffer A containing 50 mM HEPES–KOH (pH 7.6), 10 mM MgCl₂, 200 mM KCl and 25 mM imidazole. The target proteins were eluted by buffer B containing 50 mM HEPES–KOH pH 7.6, 10 mM MgCl₂, 500 mM KCl, 500 mM imidazole, and dialyzed against buffer C containing 50 mM HEPES–KOH pH 7.6, 10 mM MgCl₂, 200 mM KCl, 1 mM DTT. Protein concentrations were determined by Nanodrop (Thermo Scientific, CAT#: ND-ONE-W) and concentrated to ~4 mg/mL. For mass spectrometry quantification, protein samples were digested with trypsin and the generated peptides were analyzed using Thermofisher Q Exactive HF LC–MS/MS. Results were analyzed using maxquant 2.0.1 against a customized database that includes 34 proteins (His-tagged) in PURE system.

**In vitro translation by PURE machinery**. The 2.5× reaction buffer was composed of amino acid mixture (0.75 mM for each amino acid), 8.125 mg/mL tRNA (Roche), 5 mM ATP, 5 mM GTP, 2.5 mM CTP, 2.5 mM UTP, 50 mM creatine phosphate, 50 μg/mL folinic acid, 125 mM HEPES–KOH 7.6, 250 mM potassium glutamate, 29.5 mM magnesium acetate, 5 mM spermidine, and 2.5 mM DTT. 10× enzyme mixture was composed of 200 μg/mL creatine kinase (Roche), 300 μg/mL myokinase (Sigma-Aldrich), 50 μg/mL nucleoside 5′-diphosphate kinase from bovine liver (Sigma-Aldrich), 20 U/μL T7 RNAP (New England Biolabs), 0.002 U/μL pyrophosphatase (Thermo Fisher Scientific) and 8 U/μL recombinant RNAse inhibitor (Takara). Reactions (final volume = 10 μL) were initiated by mixing 4 μL 2.5×reaction buffer, 1 μL 10× Enzyme mixture, 15 μg MSBC-PURE, 6 μg EF-Tu, 0.1 μM ribosomes (New England Biolabs), and 50 ng of linear DNA template. After mixing, reactions were incubated at 37 °C for 4 h. The mRFP was quantified by a platereader (Biotek Synergy H1) with the excitation at 580 nm and the emission at 610 nm using a 384-well plate (Corning Spheroid Microplate).

**Consortia assembly by free cells**. BY4742(*S. cerevisiae*) constitutively expressing eGFP or MG1655 (*E. coli*) constitutively producing mCherry were pre-grown in SC or LB medium overnight. We then quantified the OD600 of the overnight culture to estimate the cell number. To create a series of consortia, we kept the total cell

number (*S. cerevisiae* and *E. coli*, ~10⁵ cells) but varied the number ratio between the two species, and inoculated the cells (in 20 μL) into 1 mL of LB + SC medium (mixing LB and SC medium in equal volume) in a 24-well plate. The plate was cultured in an incubator for 24 h at 30 °C, and the cells were analyzed by flow cytometry.

**Flow cytometry**. The cell culture (consortia assembled by free cells) was diluted by 800 times using 1× PBS (Solarbio, P1010). The sample was then analyzed by flow cytometry (BD FACSCelesta™). Fluorescence was measured for >10,000 events for each sample. We used the BD FACSDiva software for data collection and analysis. FITC-A and PI-A channels were chosen for GFP/YFP, and RFP measurement, respectively. Data were processed using FlowJo (TreeStar) to obtain the composition and fluorescence median values of different species in each sample, gated by the width of the forward scatter (FSC), the side scatter (SSC), and fluorescence channels. *S. cerevisiae* were selected with FSC-A ≥ 5.0 kV, SSC-A ≥ 5.0 kV and then screen out the cells that FITC-H ≥ 4.0 kV. *E. coli* were selected with FSC-A ≤ 5.0 kV, SSC-A ≤ 5.0 kV and then screen out the cells that PI-H ≥ 3.0 kV (Supplementary Fig. 21).

**Two-species MSBC assembly**. BY4742(*S. cerevisiae*) constitutively expressing eGFP or MG1655 (*E. coli*) constitutively producing mCherry were pre-grown in SC or LB medium overnight. The overnight culture were used to generate MSB(*S. cerevisiae*) and MSB(*E. coli*). We mixed ~1000 MSB(*S. cerevisiae*) and ~1000 MSB(*E. coli*), inoculated into 1 mL LB + SC medium and cultured at the room temperature for series photos acquisition (every 30 min) by a fluorescent microscope Olympus MVX10 (red fluorescence channel: excitation/emission = 580/610 nm; green fluorescence channel: excitation/emission = 480/520 nm; blue fluorescence channel: excitation/emission = 390/450 nm). Photos were processed using Image J to generate time-lapse videos.

We created a series consortia by tuning the seeding ratio between MSB(*S. cerevisiae*) and MSB(*E. coli*), while the total number of MSB was kept at ~2000. The MSBC were cultured in 1 mL LB + SC medium at 30 °C for 24 h. We used Image J to count the red or green capsules in the images. Assuming that each MSB has a defined carrying capacity, the GFP and mCherry areas were correlated with the cell numbers. Therefore, we calculated the ratio of *S. cerevisiae* in MSBC by dividing the number of GFP capsules by the sum of the GFP capsules and mCherry capsules, as quantified from the microscopic images (assuming the area of the capsules is the same).

**Three-species MSBC assembly**. Transparent polymethyl methacrylate (PMMA) plates (thickness = 5 mm) were cut into solid discs (as the bottom), or discs with flower-shaped chambers by the CO₂ laser cutter system (VLS 2.3, Universal Lasers, Inc., Scottsdale). The polydimethylsiloxane (PDMS) was used as the adhesive to seal the two parts and generate the culturing mold (Supplementary Fig. 9).

*E. coli* strain BL21(DE3) transformed with an inducible TagBFP plasmid, *C. glutamicum* ATCC 13032 constitutively expressing mCherry (XK99E) and *S. cerevisiae* strain BY4742 constitutively expressing eGFP were used to generate MSB(*E. coli*), MSB(*C. glutamicum*) and MSB(*S. cerevisiae*). ~400 MSB (different MSB ratio as design) were inoculated into each chamber and cultured in 200 μL LB + SC medium. 1 mM IPTG is added to induce TagBFP expression. The prepared microdevices were sealed, placed in a 30 °C incubator and cultured for up to 168 h. The MSBC were taken out at different time points for photographing by a fluorescence microscope (Olympus MVX10).

**Three-member consortia assembled by free cells**. BY4742(*S. cerevisiae*) constitutively expressing eGFP, BL21(DE3)(*E. coli*) inducibly expressing TagBFP and ATCC 13032(*C. glutamicum*) constitutively producing mCherry were pre-grown in SC or LB medium overnight. We then quantified the OD600 of the overnight culture to estimate the cell number. We inoculated the cells (1:1:1 in cell number, 20 μL in total) into 1 mL of LB + SC medium. The plate was cultured in an incubator for 24 h at 30 °C, and the cells were collected and analyzed by flow cytometry (*E. coli* (TagBFP) used Pacific Blue channel, Pacific Blue-A ≥ 540 V; *C. glutamicum* (mCherry) used PE channel, PE-A ≥ 2 kV; *S. cerevisiae* (eGFP) used FITC channel, FITC-A ≥ 1.0 kV).

**Bioremediation by MSBC**. *E. coli* strain BL21(DE3) transformed with both the ePop and Bla circuits were used to generate MSB(*E. coli*). *P. pastoris* X-33-rh-PON1 (the circuit expressing rh-PON1 was linearized and integrated into the genome of *P. pastoris* strain X-33) were used to generate MSB(*P. pastoris*). We then mixed ~75,000 MSB(*E. coli*) and ~ 75,000 MSB(*P. pastoris*), and cultured the MSBC in 50 mL BMMY medium for 48 h at 30 °C (shaking speed = 200 rpm). 1 mM IPTG was added to induce Bla expression. After fermentation, the supernatant was used for protein purification. For individual culture, ~30,000 MSB(*E. coli*) or ~30,000 MSB(*P. pastoris*) were cultured in 10 mL medium for 48 h at 30 °C (shaking speed = 200 rpm).

For cell-rescue assay, after adding 5 μL purified DOL product into 200 μL LB medium supplemented with 300 μg/mL ampicillin, 1 μL overnight culture of *E. coli* DH5α were inoculated immediately in a 96-well plate. The wells were then sealed with 50 μL mineral oil to prevent evaporation. We measured the optical density of cultures (OD600) by a platereader (Biotek Epoch 2). For data analysis, the

measured OD600 values were background corrected. Experiments were done in triplicate.

For paraoxon degradation, 1 mM PAR (paraoxon) was prepared in Tris buffer (20 mM Tris-HCl, pH 8.0, containing 1 mM CaCl$_2$). In all, 5 µL purified DOL product was added to initiate the conversion process. The formation of PNP (paranitrophenol) was monitored by measuring the absorbance at 405 nm using a platereader (Biotek Epoch 2). The consumption of PAR was calculated based on the generation of PNP. Experiments were done in triplicate.

**Bla assay**. The Bla activity in the supernatant was quantified using the nitrocefin assay (Abcam). Briefly, 5 µL supernatant was diluted into 100 µL using PBS and mixed with nitrocefin to attain a final substrate concentration of 50 µM. The resulting absorbance at 490 nm was measured using a platereader (Biotek Epoch 2) as a function of time. Enzymatic activity is determined by taking the time derivative of absorbance in the initial time window, when absorbance increases linearly with time. Experiments were done in triplicate.

**Communications of MSBC and free cells**. The sender cells (E. coli) and receiver cells (S. cerevisiae) were used to generate MSB$_{sender}$ and MSB$_{receiver}$. A series of MSBC were prepared by tuning the seeding ratio between MSB$_{sender}$ and MSB$_{receiver}$. The MSBC (~2000 MSBs) were inoculated into 1 mL LB + SC medium (with ATc and arabinose), and cultured in a 24-well plate at 30 °C for 24 h. To release bacteria from the capsules, we used 2.5 M NaCl (vortexing 5 min) followed by PBS (vortexing 10 min) to destabilize the capsules. The collected cells were diluted 800 times and analyzed by the flow cytometer. The data were analyzed by FlowJo (TreeStar) as mentioned in the previous session.

The sender cells (E. coli) and receiver cells (S. cerevisiae) were pre-grown in SC or LB medium overnight. We then quantified the OD600 of the overnight culture to estimate the cell number. To create a series of consortia, we kept the total cell number (S. cerevisiae and E. coli, ~10$^5$ cells) but varied the number ratio between the two species, and inoculated the cells (in 20 µL) into 1 mL of LB + SC medium (with ATc and arabinose) in a 24-well plate. The plate was cultured in an incubator for 24 h at 30 °C, and the cells were analyzed by flow cytometry.

**Titrate the receiver cells with AHL**. The receiver cells (S. cerevisiae) were pre-grown in SC medium overnight. We inoculated 20 µL overnight culture to 1 mL of LB + SC medium containing ATc and various concentrations of AHL (0, 0.01, 0.1, 1, 2, 5, or 10 µM), and cultured the cells in a 24-well plate at 30 °C for 24 h. The collected cells were diluted 800 times and analyzed by the flow cytometer.

**Evaluation of the signal range of MSBC during the communication**. The sender cells (E. coli) and receiver cells (S. cerevisiae) were cultured overnight in LB and SC medium respectively to generate MSB$_{sender}$ and MSB$_{receiver}$. To evaluate the communication between the free cells, we dripped 20 µL overnight culture of the sender or receiver cells onto the LB + SC agar plate (containing ATc and arabinose), and kept the grown colonies with a series of distances (0, 1.2, 4.5, or 12 mm). To evaluate the communication inside MSBC, we added ~2000 MSB$_{sender}$ or MSB$_{receiver}$ to the micro-device with two culturing chambers connected by channels in different lengths (0, 10, 20, or 30 mm), respectively. The agar plate and micro-device were placed in a 30 °C incubator for 24 h. The signal intensity was captured by a fluorescence microscope (Olympus MVX10).

**Generation and culturing of MSB(S. elongatus)**. S. elongatus were pre-grown in BG-11 medium for a few days until OD750 reaches 0.4–0.6 per bottle. In all, 1 mL of pre-grown cultures was centrifuged, resuspended in 200 µL BG-11 medium, and encapsulated into 2 mL alginate solution to generate MSB(S. elongatus). ~60,000 MSB(S. elongatus) were inoculated into 15 mL BG-11 medium and cultured at 30 °C with 20% light intensity for 72 h.

**Assembly of phototrophic MSBC**. Overall, ~20,000 MSB(S. elongatus) were inoculated into 5 mL CoBG-11 and cultured for 72 h for cell growth. After inducing by 2 mM theophylline and osmatic shock by supplementing 150 mM NaCl for 48 h, MSB(E. coli) (encapsulated by alginate) were added to form the consortia with different seeding density. The phototrophic MSBC were further cultured for 96 h before the image acquisition by fluorescence microscopy (Nikon-Ti2, excitation/emission = 590/660 nm). MSB(S. elongatus) were coated with green color by Image J for better presentation.

**Quantification of sucrose yield**. Supernatants of S. elongatus PCC 7942 or MSB(S. elongatus) cultures were collected and analyzed for sucrose quantification via a colorimetric glucose–sucrose assay kit (MAK013, Sigma-Aldrich). In this assay, glucose is oxidized via glucose oxidase, resulting in a colorimetric product (absorbance at 570 nm) that is proportional to the glucose content. To measure the sucrose content, invertase was added to convert the sucrose to glucose and fructose. The free glucose was then subtracted from the total glucose to give the concentration of sucrose present.

**Modification on the surface of the capsules by a layer-by-layer coating technique**. About 10,000 MSB were washed by PBS and incubated in 1.5 mL alginate solution (0.1% (w/v)) for 15 min with gentle shaking. The capsules were collected and incubated in 1.5 mL chitosan solution (0.4% (w/v)) with gentle shaking for another 15 min. The resultant capsules were washed again with PBS and collected for further usage.

**Evaluation of the escape rate of MSBC**. E. coli strain MC4100Z1 carrying both the ePop and inducible mCherry circuits were used to generate MSB(E. coli). S. cerevisiae strain BY4742 constitutively expressing eGFP were used to generate MSB(S. cerevisiae). Both MSBs were cultured at 30 °C for 24 h. We collected both the culture supernatant and MSBs. To release bacteria from the MSBs, we used 2.5 M NaCl (vortexing 5 min) followed by PBS (vortexing 10 min) to destabilize the capsules. The escape rate is calculated by dividing the number of cells in the supernatant by the number of cells inside MSBs. For E. coli, we quantified the escape rate based on the CFU counting. For S. cerevisiae, we quantified the escape rate by the flow cytometer.

**Evaluation of longevity and performance of the MSB over time**. E. coli strain MC4100Z1 transformed with both the ePop and inducible mCherry circuits was used to generate MSB(E. coli). ~9000 MSBs were inoculated into 3 mL M9 medium, and cultured in a six-well plate at 37 °C for 8 days. Every 24 h, the supernatant was harvested, and the new medium with 1 mM IPTG were supplemented. The fluorescence of the supernatant was evaluated by a platereader.

**Long-term storage of MSBs**. BY4742(S. cerevisiae) constitutively expressing eGFP or MG1655 (E. coli) constitutively producing mCherry were pre-grown in SC or LB medium overnight. The overnight culture were used to generate MSB(S. cerevisiae) and MSB(E. coli). These MSBs were then resuspended in the culture medium supplemented with 25% glycerol and stored at −80 °C for 12 months. The frozen MSB could be used anytime by supplementing the culture medium.

**Reporting summary**. Further information on research design is available in the Nature Research Reporting Summary linked to this article.

## Data availability

Data supporting the findings of this work are available within the paper and its Supplementary Information files. A reporting summary for this Article is available as a Supplementary Information file. Proteomics data were deposited in PRIDE with the accession number PXD034417. Source data are provided with this paper.

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

## Acknowledgements
We thank Prof. Nan Li, Weijie Wang, Wenjuan Zhao and Shengkun Dai for mass spectrometry setup; Prof. Jian Li for useful suggestions in cell-free system construction; Prof. Fan Jin and Dr. Aiguo Xia for assistance in microscopy instruments setup; Prof. Hongting Tang for assistance in fermentation setup. Dr. Yuanyuan Huang and Yanling Wu for help in strain engineering. Cyanobacteria of phototrophic MSBC in Fig. 1b were generated using the materials created with BioRender.com. This study was partially supported by the National Key Research and Development Program of China (No. 2018YFA0903000 and No. 2020YFA0908100 providing equal support) (Z.D.), Shenzhen Science and Technology Program No. KQTD20180413181837372 (Z.D.), Guangdong Natural Science Funds for Distinguished Young Scholar 2022B1515020077 (Z.D.) and National Natural Science Foundation of China 32071427 (Z.D.).

## Author contributions
L.W. designed and performed the experiments, interpreted the results, and wrote the manuscript. X.Z. designed and performed the experiments, interpreted the results and revised the manuscript. C.T. designed and performed the experiments, interpreted the results, and revised the manuscript. P.L., R.Z., and J.S. performed the experiments. Y.Z., H.C., and J.M. assisted in experimental setup and data interpretation. X.S., W.Z., X.G., and X.L. assisted in the research design and experimental setup. L.Y. assisted in research design and manuscript revision. Y.C. assisted in research design, experimental design, results interpretation, and manuscript revision. Z.D. conceived the research, designed the experiments, interpreted the results, and wrote the manuscript.

## Competing interests
The authors declare no competing interests.
