## [Peer Review File · Nature Communications]

Reviewers' Comments:

Reviewer #1:

Remarks to the Author:

The manuscript presents a large collection of experimental results using polymeric microcapsules to encapsulate bacteria or yeast. The microcapsules, called microbial swarmbots (MSBs) in the manuscript, provide spatial separation for different strains growing in close proximity. Even though the different strains are spatially separated, they can still exchange chemicals and excrete products into the surroundings. The most interesting result, in my opinion, is the ability to grow a co-culture of *E. coli* and yeast (*S. cerevisiae*) and control the relative population so that the faster growing organism (*E. coli*) does not overpopulate relative to yeast (see figure 4a). The result is not surprising given that the microcapsules limit the volume available for growth for each organism, but it is a nice approach for controlling relative population levels. My major concern with the manuscript is that none of the results were particularly surprising. It has long been known that spatial separation can provide some distinct advantages for growing microbial consortia. It has long been known that polymeric membranes can provide that spatial separation. Most of the previous work that I am familiar with has employed microfluidic devices (usually with membranes) to achieve spatial segregation, and these devices have an advantage not present in MSBs -- one has some control over the extracellular environment using inflow and outflow channels. I will provide a couple references of work in this area at the bottom of the review. I would encourage the authors to develop applications where MSBs and the spatial separation that they provide is better than a microfluidic device. Beyond this major concern, I had a few other minor comments:

- (1) Abstract: the authors should acknowledge that there are many reasons beyond "spatial organization" for microbial consortia instead of "winner-takes-all." For example, some consortia are food chains and have no spatial organization.
- (2) The Introduction has an overall lack of references.
- (3) I do not understand the purpose of the 34-member consortia that produces 34 different proteins. How is the use of microcapsules of any significant difference from using separate flasks to separate the *E. coli* strains? Are the results presented significantly different from what is observed if the *E. coli* strains are not spatially separated?
- (4) The manuscript states that there are "two organizing features" that make consortia attractive (line 182) -- division of labor and communication. I would argue that there are far more than 2 features. For example, an additional features that is not mentioned is one member of the consortia removing a molecule that is toxic to another member.
- (5) The manuscript contain numerous grammatical errors and misspellings.

Wu, F., & Dekker, C. (2016). Nanofabricated structures and microfluidic devices for bacteria: from techniques to biology. *Chemical Society Reviews*, 45(2), 268-280.

Sackmann, E. K., Fulton, A. L., & Beebe, D. J. (2014). The present and future role of microfluidics in biomedical research. *Nature*, 507(7491), 181-189.

Reviewer #2:

Remarks to the Author:

The authors of this study designed so called "microbial swarmbots consortia" (MSBC). Those are encapsulated microbial cells in chitosan polymers. In doing so they were able to co-culture multiple species like *E. coli* and *S. cerevisiae* without one of them being suppressed by the overgrowth of the other. To develop functioning co-cultures is a big goal in synthetic biology, for example to study different interactions, but it is also important in the field of metabolic engineering. The authors state here that their system would be of great use, because for example multi step production pathways could be performed in one single setup since small molecules can pass the chitosan capsule. Also, some reaction chains bare the need for the use of two different species, which can be achieved using

this system. They developed different assays in order to show this, mainly using the production of fluorescence proteins.

Concluding, after some revisions, this paper could be a good addition to your journal as a next step in the direction of metabolic engineering.

Major Remarks

Supplementary Figure 1.: The authors state that the scale bar is 10 μm . However, if this is the case, *E. coli* cells should be easily able to pass the chitosan capsule, since it is roughly 1 μm in size. Did the authors mean 10 nm? Otherwise, a control experiment using the supernatant to check for escaped bacteria would be suitable. In general, it would be important to know, if there is any leakiness of the system whatsoever. Experimental evidence for this is requested.

Figure 3: What is the advantage of using MSBC in this case? Could it also be achieved by co-culturing the strains, since they are all the same species and should be able to grow in a mixed culture? An experimental advantage of the MSBC would be interesting.

Minor Remarks

The authors should revise grammar and language in some sentences.

In general, the figure captions should be revised, since they should be understandable without reading the text. This is not the case for most figures.

Figure 5b: Could the authors comment on the impurities seen in the third blot? Since it is stated that also this was purified as the single samples. Also, is there an explanation why the production of hr-PON1 seems to be much higher than for Bla?

Figure 6f: What are the red arrows indicating? Also, for me, it is not that easy to see, that *E. coli* grew using the higher ratio. Could this experiment be repeated by using for example the mCherry cells? It is also not visible if the *E. coli* are growing within the Chitosan capsule or not, when interpreting the darker spots as cells.

Supplementary Figure 3: What are the concentrations used for the standards?

Supplementary Figure 11: It would be advisable to label the figures a-c and then explain each panel.

Supplemental Video 1: Could the authors comment on why the *E. coli* (red) are spotted and not very dense in this video, especially in comparison to Figure 2a? The yeast (green) seems to be able to colonize the whole capsules?

Reviewer #3:

Remarks to the Author:

This manuscript seeks to address the problem of interspecies nutrient competition (and the resulting potential for overgrowth by certain species) within engineered microbial consortia. Specifically, the authors encapsulate microbial subpopulations in polymeric microcapsules (termed microbial swarmbots, or MSBs) and crosslink them to form spatially segregated, multi-species synthetic communities (termed microbial swarmbot consortia, or MSBC). This strategy ensures that the maximum volume allocated to a given microbial subpopulation never changes, since species are confined to their respective microcapsules. The potential advantage of this approach is the ability to assemble various synthetic microbial communities in a modular fashion while guaranteeing that no individual species takes over the population, throwing the community balance out of proportion. The authors illustrate the proof of concept for this approach using several examples in biomanufacturing, bioremediation, and engineered photosynthesis.

Overall, I think this a solid manuscript that builds on prior efforts (by the authors as well as others) to develop microbial swarmbots that will be of interest to those in the synthetic biology community. The novelty here lies in the number and variety of illustrative proof-of-concept MSBC examples, including some with large numbers of individual MSBs. Since a key challenge in developing any synthetic biology tool is to demonstrate its robustness and utility in a variety of contexts, I think this is a valuable contribution to the field. It would further strengthen the manuscript to (wherever possible, but especially Figures 4, 5, and 6) compare the MSBC approach to unencapsulated controls. Before publication I would suggest a close read through for writing and grammar, especially in the

introduction and discussion. I provide a few specific questions and comments below, but overall am relatively enthusiastic for publication.

Specific Comments:

1) In Figure 3 the authors develop a 34 member MSBC to enable assembly of the PURE expression system. This is valuable as a proof-of-concept demonstrating the ability of assembling such a large number of MSBs, but is there some advantage to doing it this way (vs. in an individual cell)? An overall question I had with this manuscript is when the MSBC approach is actually more useful than existing methods. I am onboard with the future potential of the MSBC approach but I think it would help the reader if any near-term advantages are clearly identified. Further, since cells are auto-lysing, are the lysates contained in MSBs? How long can this go before the MSB microenvironment is too toxic for cell growth? Is this compatible with production and expression needs more generally?

2) In Figure 4 the authors pattern individual MSBs into various shapes. Showing long term stability of MSBC patterns over time would be very nice here, especially as compared to unencapsulated controls with the same initial distribution. Won't cells just run out of space and stop growing rather than disrupting the pattern?

3) In Figure 5, the authors generate an MSBC composed of senders of receivers, forming a division of labor. While control over strength of signal is demonstrated, what about spatial range? It would be nice to see some different spatial arrangements to test how far senders and receivers in MSBC can be from each other amidst other cells (non-participants). This could form the basis of demonstrating an advantage for the MSBC approach, since overgrowth might push senders or receivers too far apart from optimal separation.

4) In general, it would strengthen the manuscript to (wherever possible, but especially Figures 4, 5, and 6) compare the MSBC approach to unencapsulated controls to demonstrate an advantage.

Reviewer #4:

Remarks to the Author:

This study aims to build a stable consortium by developing a microbial swarmbot-mediated spatial segregation strategy. This research belongs to the new frontier study for synthetic biology. The findings and results could address the challenges in the buildup studies of artificial microbial communities. The technology established in this study could have potential in many useful research fields. As currently presented, there are still a few points that need to be addressed.

1. What is the expected maximum protein size that can be released from the MSB's?
2. How long can the system operate since the growth of the cell and the cell debris may eventually accumulate to problematic levels.
3. Can the authors compare the yield of CBGA between the MSB and the free culture?
4. I have noticed that in Figure 4 and figure 5, the authors used different methods to quantify the populations of the different species (microscopy and flow cytometry). What is the reason for it?
5. In figure 4, the authors demonstrated the assembly of the three species consortia. While the results are persuasive, the authors should comment on the scalability of the system since it was done on the microfluidic chips.
6. It is very intuitive to construct the MSBC phototrophic community. Can the authors compare the yield of sucrose between the MSB (*S. elongates*) and the free culture?

Below, we provide point-by-point responses to reviewers' comments (*italicized, blue*). The revision in the manuscript is highlighted in blue.

Referee #1: The manuscript presents a large collection of experimental results using polymeric microcapsules to encapsulate bacteria or yeast. The microcapsules, called microbial swarmbots (MSBs) in the manuscript, provide spatial separation for different strains growing in close proximity. Even though the different strains are spatially separated, they can still exchange chemicals and excrete products into the surroundings. The most interesting result, in my opinion, is the ability to grow a co-culture of E. coli and yeast (S. cerevisiae) and control the relative population so that the faster growing organism (E. coli) does not overpopulate relative to yeast (see figure 4a). The result is not surprising given that the microcapsules limit the volume available for growth for each organism, but it is a nice approach for controlling relative population levels.

We thank the reviewer for recognizing the values of our study and for providing insightful comments.

My major concern with the manuscript is that none of the results were particularly surprising. It has long been known that spatial separation can provide some distinct advantages for growing microbial consortia. It has long been known that polymeric membranes can provide that spatial separation. Most of the previous work that I am familiar with has employed microfluidic devices (usually with membranes) to achieve spatial segregation, and these devices have an advantage not present in MSBs -- one has some control over the extracellular environment using inflow and outflow channels. I will provide a couple references of work in this area at the bottom of the review. I would encourage the authors to develop applications where MSBs and the spatial separation that they provide is better than a microfluidic device.

We thank the reviewer for evaluating our manuscript. These comments reveal multiple aspects that we should further clarify, including the conceptual novelty, approach, scope, and implications.

In this work, we developed a microcapsule-based spatial segregation strategy to culture microbial consortia composed of single species to multispecies. We applied this strategy to a range of applications, including biomanufacturing and bioremediation *at different scales (from microliters to liters)*. The reviewer is absolutely correct that it has been known that spatial separation can provide some distinct advantages for growing microbial consortia. However, as the reviewer mentioned, most studies to date utilized membrane-embedded or specifically designed (e.g. fabrication of hundred nanometers deep nanoslits which allow for the diffusion of nutrients, metabolites and signaling molecules while being too shallow for bacteria to pass through) microfluidic chips to investigate the spatial organization of the microbial consortia¹⁻⁶. *The design of these chips is very intuitive, but the strategies are generally not scalable for synthetic biology implementations, due to the limited scale (usually below 1 mL for microfluidics)^{7,8}, the complicated fabrication and operation processes (such as the fabrication of membrane-embedded chips)^{2,9,10}, and the lack of the modularity (have to frequently modify the design based on the type of the consortia and the aims)^{1,3,4}.*

As such, our work makes several conceptual and technological advances.

1. Our work demonstrates the use of **the engineered microbes** and **biomaterials** to assemble the microbial community consisting of single species or multi-species. The microcapsule has a three-dimensional crosslinked structure, which allows the transport of proteins and small molecules including nutrients, signaling molecules and metabolites, but traps the microbes and fences the subpopulations. Our study provides a unified and intrinsic mechanism to maintain the stability of the engineered communities, offering **a generalized platform** for microbial consortia construction.
2. This work demonstrates **a new, feasible and scalable approach** to control the consortia composition with precision.
3. By design, the MSBC strategy is **modular**. The consortia can be built with any MSBs containing different strains or species based on the specific design. The host microbes and the encapsulating material can be separately engineered or optimized and then integrated. These properties make our platform highly versatile and flexible.

- Using this strategy, we have demonstrated **a series of applications at different scales**, including the division of labor within the same species (manufacturing 34-enzyme system, in liters) and across the species (degrading agriculture wastewater, in hundreds of milliliters), the assembly of a phototrophic microbial community (in milliliters) and the communication across the species (in hundreds of microliters). *Collectively*, these examples prove that MSBC platform is potentially effective for assembling a wide range of consortia and achieving the diverse purposes.

Beyond this major concern, I had a few other minor comments:

- Abstract: the authors should acknowledge that there are many reasons beyond "spatial organization" for microbial consortia instead of "winner-takes-all." For example, some consortia are food chains and have no spatial organization.*

The reviewer is correct that multiple mechanisms can allow the maintenance of microbial consortia. In addition to spatial separation, maintaining a proper interaction network among members can also allow co-existence. Spatial organization specifically addresses the challenges arising from “winner-takes-all”, where a fast-growing population overtakes slow-growing populations (in the absence of countering mechanisms).

We have updated the main text to better describe the rationale.

- The Introduction has an overall lack of references.*

We thank the reviewer for this important point. In terms of the reviewer’s comment, we have supplemented more references about 1) using microfluidics to investigate the spatial organization, and 2) programming the microbial consortia through symbiosis in the main text.

- I do not understand the purpose of the 34-member consortia that produces 34 different proteins. How is the use of microcapsules of any significant difference from using separate flasks to separate the E. coli strains? Are the results presented significantly different from what is observed if the E. coli strains are not spatially separated?*

We thank the reviewer for this important point. One merit of microbial consortia is the division of labor, in which the community can conduct the complex functions that cannot be achieved by the individual members. Compared with culturing individual strains in separate flasks, using microbial consortia to produce the multi-enzyme systems in one-pot (especially for 34-enzyme included systems) largely decreases the efforts and time during the culturing, cell harvest and also the downstream processes.

In the particular case of 34-member MSBC, we encapsulated the cells (within the same species and with a similar growth rate) to assemble the microbial community. Therefore, simply mixing these cells can also assemble stable consortium for PURE production. Nevertheless, we used this example to demonstrate **three features** of the MSBC system.

- The MSBC system can **integrate the lysis and separation** steps during the biomanufacturing¹¹. For each individual MSB, by sensing the confinement, the engineered bacteria undergo programmed partial lysis at a high local density. The released proteins were transported to the exterior of the capsules, while the cell factories were trapped (**Supplementary Figure 1 and 2**). **The encapsulation is a critical component of the system design to facilitate protein isolation in a semi-continuous manner (Supplementary figure 7)**. These traits together with the one-pot synthesis made the biomanufacturing using MSBC more accessible and portable. We have added the **Supplementary Figure 2 and 7** to better explain this point.
- A key challenge in developing any synthetic biology tool is to demonstrate its **robustness**. As a supplement to the multi-species consortia assembly, the assembly of 34-member MSBC (within the same species) proves the robustness of MSBC platform in both making 34 functional individual MSBs and building the MSBC with the vast number of subpopulations to attain a desired function.

3. This example demonstrated the **broad utility** of the MSBC. We used MSBC in this example to produce PURE, which is **a representative complex and complicated 34-enzyme system** and has enormous potential in biochemistry and bioengineering. The success in producing PURE indicated that the MSBC is potentially applicable to manufacture a broad range of other multi-enzyme systems for the synthesis of both biomacromolecules and value-added natural products.

We have further elaborated these points in the revised context.

- *The manuscript states that there are "two organizing features" that make consortia attractive (line 182) -- division of labor and communication. I would argue that there are far more than 2 features. For example, an additional features that is not mentioned is one member of the consortia removing a molecule that is toxic to another member.*

The reviewer is correct that there are additional advantages of microbial consortia beyond division of labor and communication. We have revised our writing to be more rigorous, by also citing references along the line of the example that the reviewer gave.

- *The manuscript contain numerous grammatical errors and misspellings.*

We thank the reviewer for the careful examination. We have revised the whole package to improve the language and eliminate the grammar errors.

We thank the reviewer again for providing the insightful comments.

Referee #2 (Remarks to the Author):

*The authors of this study designed so called “microbial swarmlots consortia” (MSBC). Those are encapsulated microbial cells in chitosan polymers. In doing so they were able to co-culture multiple species like *E. coli* and *S. cerevisiae* without one of them being suppressed by the overgrowth of the other. To develop functioning co-cultures is a big goal in synthetic biology, for example to study different interactions, but it is also important in the field of metabolic engineering. The authors state here that their system would be of great use, because for example multi step production pathways could be performed in one single setup since small molecules can pass the chitosan capsule. Also, some reaction chains bare the need for the use of two different species, which can be achieved using this system. They developed different assays in order to show this, mainly using the production of fluorescence proteins. Concluding, after some revisions, this paper could be a good addition to your journal as a next step in the direction of metabolic engineering.*

We thank the reviewer for recognizing the novelty and significance of our study and for providing insightful comments.

Major Remarks

• *Supplementary Figure 1.: The authors state that the scale bar is 10 μm . However, if this is the case, *E. coli* cells should be easily able to pass the chitosan capsule, since it is roughly 1 μm in size. Did the authors mean 10 nm? Otherwise, a control experiment using the supernatant to check for escaped bacteria would be suitable. In general, it would be important to know, if there is any leakiness of the system whatsoever. Experimental evidence for this is requested.*

We thank the reviewer for the careful examination and the important note. The error bar of the image is correct. In this original SEM image, we dissected the capsule into the thin layer to better present the pore structure. Although some of the pore size in the microcapsule is larger than the size of the *E. coli* ($\sim 1 \mu\text{m}$), these pores were crosslinked in three-dimensional, rather than just a single-layer mesh. In terms of the reviewer’s comment and to avoid the confusion, we have supplemented additional SEM images to show the three-dimensional crosslinked structure of the capsules, as shown in **Supplementary Figure 1**.

Given the reviewer’s comments, we have quantified the escape rate of bacteria or yeast from capsules (number of the cells in the supernatant/total cell number in capsules) in **Supplementary Figure 19**. After 24 hours culture, the escape rate for *E. coli* or *S. cerevisiae* were $\sim 0.2\%$ and 0.003% , respectively. The escape rate can be modulated by a surface coating technique. We first coated the capsules with a layer of alginate, followed by another layer of chitosan. After this layer-by-layer modification, the escape rate of the MSB(*E. coli*) decreased to zero, after 24 hours of culturing. We have further elaborated this part in the revised text.

• *Figure 3: What is the advantage of using MSBC in this case? Could it also be achieved by co-culturing the strains, since they are all the same species and should be able to grow in a mixed culture? An experimental advantage of the MSBC would be interesting.*

We thank the reviewer for this important point. For the 34-member MSBC, we encapsulated the cells (within same species and with a similar growth rate) to assemble the microbial community. The reviewer is correct about that simply mixing these cells can also assemble a stable consortium for PURE production. Nevertheless, we used this example to demonstrate **three features** of the MSBC system.

1. The MSBC system can **integrate the lysis and separation** steps during the biomanufacturing¹¹. For each individual MSB, by sensing the confinement, the engineered bacteria undergo programmed partial lysis at a high local density. The released proteins were transported to the exterior of the capsules, while the cell factories were trapped (**Supplementary Figure 1 and 2**). **The encapsulation is a critical component of the system design to facilitate protein isolation in a semi-continuous manner**. These traits together with the one-pot synthesis made the biomanufacturing using MSBC more accessible and portable. We have added the **Supplementary Figure 2 and 7** to better explain this point.

2. A key challenge in developing any synthetic biology tool is to demonstrate its **robustness**. As a supplement to the multi-species consortia assembly, the assembly of 34-member MSBC (within the same species) proves the robustness of MSBC platform in both making 34 functional individual MSBs and also building the MSBC with a vast number of subpopulations to attain a desired function.
3. This example demonstrated the **broad utility** of the MSBC. We used MSBC in this example to produce PURE, which is **a representative complex and complicated 34-enzyme system** and has enormous potential in biochemistry and bioengineering. The success in producing PURE indicated that the MSBC is potentially applicable to manufacture a broad range of other multi-enzyme systems for the synthesis of both biomacromolecules and value-added natural products.

We have further elaborated these points in the revised context.

Minor Remarks

- *The authors should revise grammar and language in some sentences.*

We thank the reviewer for the careful examination. We have revised the whole package to improve the language and eliminate the grammar errors.

- *In general, the figure captions should be revised, since they should be understandable without reading the text. This is not the case for most figures.*

We thank the reviewer for the comments. We have revised the figure captions to make them self-explanatory without reading the main text.

- *Figure 5b: Could the authors comment on the impurities seen in the third blot? Since it is stated that also this was purified as the single samples. Also, is there an explanation why the production of hr-PON1 seems to be much higher than for Bla?*

We thank the reviewer for the comment. There are two reasons that may account for the impurities in the third blot. First, the DOL products were harvested from the larger scale cell culture. We cultured 10 mL MSB(*E. coli*) or 10 mL MSB(*P. pastoris*) to produce the β -lactamase or hr-PON1, respectively. For the DOL product, the sample were collected and purified from 50 mL MSBC. The concentration of the DOL sample (running the SDS-PAGE) is higher than that of the individual cultured MSB samples, causing the impurities shown on the gel. The other possible reason is the crossfeeding between the *E. coli* and *P. pastoris* in MSBC. Previous research reported that the yeast overflowed the amino acids that enabled survival of the lactic acid bacteria¹². The crossfeeding might promote the growth of *E. coli* and yielded more protein (including the impurities). We have further elaborated these points in both the main text and the method.

To explain the difference in the yield, one possible reason is the difference in the molecular weight of the protein. Comparative studies have shown that the probability of soluble expression in *E. coli* decreases with the molecular weight increasing, especially for proteins > 60 kD¹³. To stabilize the enzyme, we fused an elastin like polypeptides (ELPs) with the Bla, and the resultant molecular weight is ~ 60 kDa. In comparison, the molecular weight of hr-PON1 is ~ 40 kDa.

- *Figure 6f: What are the red arrows indicating? Also, for me, it is not that easy to see, that E. coli grew using the higher ratio. Could this experiment be repeated by using for example the mCherry cells? It is also not visible if the E. coli are growing within the Chitosan capsule or not, when interpreting the darker spots as cells.*

We thank the reviewer for the comment. It is generally reported that *S. elongatus* have emissions at 650-680 nm by excitation^{14,15}. In our experimental setting, we excited the MSBC at 590 nm, and collected the emission at 662 nm. We applied the green color to coat the MSB(*S. elongatus*) for better presentation since the natural MSB(*S. elongatus*) showed a green color (**Figure 6b**). We have updated the method to elaborate this part.

In terms of the reviewer's question, we used GFP labeled *E. coli* (sucrose-utilizing strain) to repeat the experiments. Although the colonies of *E. coli* grew in the capsules, there was hardly any fluorescence signal detected. To troubleshoot the reason, we inoculated the free cells (with GFP plasmid) into LB or CoBG11 supplemented with 0.2% sucrose, and cultured the cells for 24 hrs. Cells grew in both media. However, the GFP signal in LB cultured cells was ~580 times higher than that CoBG11 with 0.2% sucrose cultured cells (Figure 1). In fact, the GFP signal of cells carrying the GFP plasmid was almost equivalent to the signal of cells not carrying the plasmid, both were cultured in CoBG11 with 0.2% sucrose.

We measured the pH of the culture (free cells carrying the GFP plasmid in LB or CoBG11 with 0.2% sucrose), and found the pH was ~ 7.5 for LB culture and ~ 5.0 for CoBG11 with 0.2% sucrose. Therefore, we concluded that the low pH of the culturing environment caused the diminishing of the GFP. It is reported widely that both GFP and YFP are sensitive to the pH ($pK_a = 5.5 - 6$) and do not glow under acidic condition¹⁶⁻¹⁸. We also tried the YFP labeled cells and found the similar phenomena.

Figure 1. Comparison of GFP intensity of cells cultured in different media.

To better present the *E. coli* colonies and to prove these colonies were inside the capsules, we used the *E. coli* (sucrose-utilizing strain) to generate the MSB(*E. coli*). By culturing these MSB in the CoBG11 with 0.2% sucrose for 96 hrs, colonies appeared inside the MSB (**Supplementary Figure 17**). We also supplemented multiple magnified images of co-cultured MSBC (only CoBG11 without any organic carbon source) to better present the *E. coli* colonies inside MSBC (**Figure 6e-f** and **Supplementary Figure 18**). The growth state of the *E. coli* in MSBC was resembled with the MSB(*E. coli*) supplemented with the sucrose, underlying MSB_{photoautotroph} could produce sucrose from CO₂ and supported the growth of the MSB_{heterotroph}. We have revised the package to better present the figures.

- *Supplementary Figure 3: What are the concentrations used for the standards?*

We thank the reviewer for the comment. The concentration for the standards (both OA and CBGA) is 50 μ M. We have supplemented this information in the figure caption.

- *Supplementary Figure 11: It would be advisable to label the figures a-c and then explain each panel.*

We thank the reviewer for the comment. We have labeled the panels individually to better present the figure and explained each panel.

- *Supplemental Video 1: Could the authors comment on why the E. coli (red) are spotted and not very dense in this video, especially in comparison to Figure 2a? The yeast (green) seems to be able to colonize the whole capsules?*

We thank the reviewer for the comment. The MSBC were cultured at the room temperature (~ 22 °C) on the fluorescence microscopy for series photo acquisition (**Supplementary Video 1**). In **Figure 2 and 4**, the

individual MSB or MSBC were cultured at 30 °C in the incubator. We have elaborated these details in the captions of related figure and video.

We thank the reviewer again for providing the insightful comments.

Referee #3 (Remarks to the Author):

This manuscript seeks to address the problem of interspecies nutrient competition (and the resulting potential for overgrowth by certain species) within engineered microbial consortia. Specifically, the authors encapsulate microbial subpopulations in polymeric microcapsules (termed microbial swarmbots, or MSBs) and crosslink them to form spatially segregated, multi-species synthetic communities (termed microbial swarmbot consortia, or MSBC). This strategy ensures that the maximum volume allocated to a given microbial subpopulation never changes, since species are confined to their respective microcapsules. The potential advantage of this approach is the ability to assemble various synthetic microbial communities in a modular fashion while guaranteeing that no individual species takes over the population, throwing the community balance out of proportion. The authors illustrate the proof of concept for this approach using several examples in biomanufacturing, bioremediation, and engineered photosynthesis.

Overall, I think this a solid manuscript that builds on prior efforts (by the authors as well as others) to develop microbial swarmbots that will be of interest to those in the synthetic biology community. The novelty here lies in the number and variety of illustrative proof-of-concept MSBC examples, including some with large numbers of individual MSBs. Since a key challenge in developing any synthetic biology tool is to demonstrate its robustness and utility in a variety of contexts, I think this is a valuable contribution to the field. It would further strengthen the manuscript to (wherever possible, but especially Figures 4, 5, and 6) compare the MSBC approach to unencapsulated controls. Before publication I would suggest a close read through for writing and grammar, especially in the introduction and discussion. I provide a few specific questions and comments below, but overall am relatively enthusiastic for publication.

We thank the reviewer for recognizing the novelty and significance of our study and for providing insightful comments.

Specific Comments:

• In Figure 3 the authors develop a 34 member MSBC to enable assembly of the PURE expression system. This is valuable as a proof-of-concept demonstrating the ability of assembling such a large number of MSBs, but is there some advantage to doing it this way (vs. in an individual cell)? An overall question I had with this manuscript is when the MSBC approach is actually more useful than existing methods. I am onboard with the future potential of the MSBC approach but I think it would help the reader if any near-term advantages are clearly identified.

We thank the reviewer for this important point. For the 34-member MSBC, we encapsulated the cells (within same species and with a similar growth rate) to assemble the consortia. We used this example to demonstrate **three features** of the MSBC system.

1. The MSBC system can **integrate the lysis and separation** steps during the biomanufacturing¹¹. For each individual MSB, by sensing the confinement, the engineered bacteria undergo programmed partial lysis at a high local density. The released proteins were transported to the exterior of the capsules, while the cell factories were trapped (**Supplementary Figure 1 and 2**). **The encapsulation is a critical component of the system design to facilitate protein isolation in a semi-continuous manner.** These traits together with the one-pot synthesis made the biomanufacturing using MSBC more accessible and portable. We have added the **Supplementary Figure 2 and 7** to better explain this point.
2. A key challenge in developing any synthetic biology tool is to demonstrate its **robustness**. As a supplement to the multi-species consortia assembly, the assembly of 34-member MSBC (within the same species) proves the robustness of MSBC platform in both making 34 functional individual MSBs and building the MSBC with the vast number of subpopulations to attain a desired function.
3. This example demonstrated the **broad utility** of the MSBC. We used MSBC in this example to produce PURE, which is a **representative complex and complicated 34-enzyme system** and has enormous potential in biochemistry and bioengineering. The success in producing PURE indicated that the MSBC is potentially applicable to manufacture a broad range of other multi-enzyme systems for the synthesis of both biomacromolecules and value-added natural products.

We have further elaborated these points in the revised context.

• Further, since cells are auto-lysing, are the lysates contained in MSBs? How long can this go before the MSB microenvironment is too toxic for cell growth? Is this compatible with production and expression needs more generally?

We thank the reviewer for the comments. The porous structure allows the exchange between metabolites and nutrients between the interior and exterior of the capsules. In our original submission, we have maintained protein production for up to 96 hrs for the manufacturing of DOL product of MSBC (**Figure 5**).

To further examine the long-term performance of the system, we have conducted additional experiments to examine a longer-term operation. We fabricated MSBs encapsulating cells that could grow and autolysis to release the fluorescent protein (MC4100Z1(ePop/mCherry)), and cultured the MSBs in M9 medium. The supernatant was collected every 24 hrs, and the MSBs were washed by PBS and supplemented with new nutrients. The mCherry in the supernatant was quantified by a platerreader. Our results showed that the MSB could be reused, by periodically replacing the growth media for eight days (192 hrs, **Supplementary Figure 19b**). The protein yield was maintained at ~30% of the peak value (on day 3); they started to disintegrate after one week.

• In Figure 4 the authors pattern individual MSBs into various shapes. Showing long term stability of MSBC patterns over time would be very nice here, especially as compared to unencapsulated controls with the same initial distribution. Won't cells just run out of space and stop growing rather than disrupting the pattern?

We thank the reviewer for the comments. Given the reviewer's comment, we first evaluated the consortia assembled by the un-encapsulated controls. Homogenous culture of the *E. coli*, *S. cerevisiae* and *C. glutamicum* (seeding ratio = 1:1:1) led to the domination of one species at the end (**Supplementary Figure 10a**). In comparison, MSBC could stabilize the consortia with each species actively growing within the boundary.

We have also supplemented the additional experiments to evaluate the long term stability of the MSBC pattern. The MSBC were assembled by mixing the MSB(*E. coli*), MSB(*S. cerevisiae*) and MSB(*C. glutamicum*). Our results showed that the MSBC patterns were well maintained even after 7 days, underscoring the stability of the system (**Supplementary Figure 10b**).

• In Figure 5, the authors generate an MSBC composed of senders of receivers, forming a division of labor. While control over strength of signal is demonstrated, what about spatial range? It would be nice to see some different spatial arrangements to test how far senders and receivers in MSBC can be from each other amidst other cells (non-participants). This could form the basis of demonstrating an advantage for the MSBC approach, since overgrowth might push senders or receivers too far apart from optimal separation.

We thank the reviewer for the inspiring comments. As the reviewer mentioned, one merit of the MSBC platform is that the spatial range of signal in the communication can be modulated in a flexible way, since the separation of the cells were based on the polymeric capsules. To this notion, we fabricated several micro-device with two chambers connected by channels in different lengths. We inoculated the MSB_{sender} and MSB_{receiver} into the chambers separately. Results showed that the cells of MSB_{receiver} were activated at different spatial ranges, and the fluorescence of the MSB_{receiver} was homogenous (**Supplementary Figure 15a**).

In contrast, it is not easy to create the spatial range in well-mixed consortia. One method for separation is to inoculate the sender and receiver cells on the agar plates with different distances. However, the fluorescence of the receiver was not homogenous since the growth of the sender and receiver cells might push the cells far apart. Increasing the spatial range of the sender and receiver cells could let the cells grow independently without the interference. However, the signal quickly deactivated as shown by the diminished fluorescence of the receiver cells (**Supplementary Figure 15b**). We have further elaborated these details in the revised text.

• In general, it would strengthen the manuscript to (wherever possible, but especially Figures 4, 5, and 6) compare the MSBC approach to unencapsulated controls to demonstrate an advantage.

We thank the reviewer for the comments. In terms of the reviewer's comment, we have supplemented the additional experiments to demonstrate the advantage of the MSBC approach as compared with the unencapsulated controls. We summarized these details in below.

- 1) In figure 4, we further analyzed the composition of the consortia made by simply mixing the three species (**Supplementary Figure 10a**). We inoculated the *E. coli* (BFP tagged), *S. cerevisiae* (GFP tagged) and *C. glutamicum* (mCherry tagged) into the medium (inoculation ratio = 1:1:1), cultured the well-mixed consortium for 24 hours and analyzed the composition by flow cytometer. The results showed that *E. coli* persisted in the community at the end for all replicates. In comparison, the MSBC approach can control the species ratio with precision (**Figure 4e-f**). Our additional experiment also showed that MSBC approach maintained the stability of the consortia even after 7 days (**Supplementary Figure 10b**).
- 2) In the original figure 5, we demonstrated the advantage in the precise control of signal strength by the MSBC approach. We further conducted experiments to compare the control in the signal range in communication by MSBC or free cells. Our result showed that the MSBC platform could modulate the spatial range of communication in a flexible way. The fluorescence of the MSB_{receiver} was activated at different spatial ranges with a homogenous expression among the receiver cells (**Supplementary Figure 15a**). In contrast, it is not easy to create the spatial range in a well-mixed consortia. The fluorescence of the receiver in well-mixed consortia was also not homogenous since the growth of the sender and receiver cells might push the cells far apart. Increasing the spatial range of the sender and receiver cells could let the cells grow independently without the interference. However, the signal quickly deactivated as shown by the diminished fluorescence of the receiver cells (**Supplementary Figure 15b**).
- 3) Compared with the free-cell suspension culture, the immobilization of the cyanobacteria could potentially increase its viability and metabolic activity, as reported by multiple previous research¹⁹⁻²¹. In our experimental setup, the sucrose yield of MSB(*S. elongatus*) was ~ 73% higher than that of the free *S. elongatus* (**Supplementary Figure 16**). We have further elaborated this part in the revised context.

We thank the reviewer again for providing the insightful comments.

Referee #4 (Remarks to the Author):

This study aims to build a stable consortium by developing a microbial swarmbot-mediated spatial segregation strategy. This research belongs to the new frontier study for synthetic biology. The findings and results could address the challenges in the buildup studies of artificial microbial communities. The technology established in this study could have potential in many useful research fields. As currently presented, there are still a few points that need to be addressed.

We thank the reviewer for recognizing the novelty and significance of our study and for providing insightful comments.

- *What is the expected maximum protein size that can be released from the MSB's?*

We thank the reviewer for this question. By far, the largest protein we have tested is around 100kDa. In terms of the reviewer's comment, we have further used a model polymer to evaluate the transport capacity of the MSB. We encapsulated a rhodamine labeled polymer (~150kDa) by polymeric capsules, cultured the capsules in the medium and measured the fluorescence in the supernatant. Our data showed a gradual increase in the fluorescence signal (**Supplementary Figure 2**), indicating that the polymer can be effectively transported out of the capsules.

- *How long can the system operate since the growth of the cell and the cell debris may eventually accumulate to problematic levels.*

We thank the reviewer for the comments. The porous structure allows the exchange between metabolites and nutrients between the interior and exterior of the capsules. In our original submission, we have maintained protein production for up to 96 hrs for the manufacturing of DOL product of MSBC (**Figure 5**).

To further examine the long-term performance of the system, we conducted an additional experiment to examine a longer-term operation. We fabricated MSBs encapsulating cells that could grow and autolysis to release the fluorescent protein (MC4100Z1(ePop/mCherry)), and cultured the MSBs in M9 medium. The supernatant was collected every 24 hrs, and the MSBs were washed by PBS and supplemented with new nutrients. The mCherry in the supernatant was quantified by a platerader. Our results showed that the MSB could be reused, by periodically replacing the growth media for eight days (192 hrs, **Supplementary Figure 19b**). The protein yield was maintained at ~30% of the peak value (on day 3); they started to disintegrate after one week.

- *Can the authors compare the yield of CBGA between the MSB and the free culture?*

We thank the reviewer for the constructive comment. Given the reviewer's comment, we have further compared the yield of CBGA between the engineered *S. cerevisiae* and MSB(*S. cerevisiae*). Based on our results, The yield of CBGA by MSB(*S. cerevisiae*) was comparable with that of the free cells (**Supplementary Figure 6**), underscoring the metabolic activity of the cells inside the MSB.

- *I have noticed that in Figure 4 and figure 5, the authors used different methods to quantify the populations of the different species (microscopy and flow cytometry). What is the reason for it?*

We thank the reviewer for the important question. In figure 4, we cultured MSBC and tracked its growth by microscopy. We quantified the subpopulations of MSBC assuming that each MSB had a defined carrying capacity and the number of the capsules was correlated with the cell number. We calculated the *S. cerevisiae* to *E. coli* ratio by dividing the number of GFP capsules by that of the mCherry from the microscopic images (assuming the area of the capsules is the same).

In figure 5, we need to release the cells from the MSBC since measuring the fluorescence intensity of the receiver cells was required to evaluate the signal transduction in the communication. Therefore, we further used the flow cytometry data to quantify the subpopulations of the different species. The cell counting results

by flow cytometry were in full accordance with the quantification by microscopy, indicating that the MSBC method was capable of controlling the subpopulation ratio with precision.

• In figure 4, the authors demonstrated the assembly of the three species consortia. While the results are persuasive, the authors should comment on the scalability of the system since it was done on the microfluidic chips.

We thank the reviewer for the constructive comment. One merit of the MSBC system is its scalability. In the current work, we have demonstrated a series of applications at different scales, including the division of labor within the same species (manufacturing 34-enzyme system, in liters) and across the species (degrading agriculture wastewater, in hundreds of milliliters), the communication across the species (in hundreds of microliters), and the assembly of a phototrophic microbial community (in milliliters). The current experimental setting for the **Figure 4** is in several hundred micron-liter, and it can be readily scaled up to milliliters or even liters by increasing the amount of the MSB.

*• It is very intuitive to construct the MSBC phototrophic community. Can the authors compare the yield of sucrose between the MSB (*S. elongatus*) and the free culture?*

We thank the reviewer for the constructive comment. Compared with the free cell suspension culture, the immobilization of the cyanobacteria could potentially increase its viability and metabolic activity, as reported by multiple previous research¹⁹⁻²¹. In terms of the reviewer's comment, we have further compared the yield of sucrose between the MSB(*S. elongatus*) and the free culture. Compared with the free cells, MSB(*S. elongatus*) has a $\sim 73\%$ increase in the yield of the sucrose (**Supplementary Figure 15**). This is possibly caused by an enhanced metabolic activity of *S. elongatus* by immobilization¹⁹⁻²¹.

We thank the reviewer again for providing the insightful comments.

Reference

- 1 Kim, H. J., Boedicker, J. Q., Choi, J. W. & Ismagilov, R. F. Defined spatial structure stabilizes a synthetic multispecies bacterial community. *Proceedings of the National Academy of Sciences of the United States of America* **105**, 18188-18193, (2008).
- 2 Wu, F. B. & Dekker, C. Nanofabricated structures and microfluidic devices for bacteria: from techniques to biology. *Chem Soc Rev* **45**, 268-280, (2016).
- 3 Boedicker, J. Q., Vincent, M. E. & Ismagilov, R. F. Microfluidic Confinement of Single Cells of Bacteria in Small Volumes Initiates High-Density Behavior of Quorum Sensing and Growth and Reveals Its Variability. *Angewandte Chemie-International Edition* **48**, 5908-5911, (2009).
- 4 Luo, X. L., Tsao, C. Y., Wu, H. C., Quan, D. N., Payne, G. F., Rubloff, G. W. & Bentley, W. E. Distal modulation of bacterial cell-cell signalling in a synthetic ecosystem using partitioned microfluidics. *Lab Chip Lab Chip* **15**, 1842-1851, (2015).
- 5 van Vliet, S., Hol, F. J. H., Weenink, T., Galajda, P. & Keymer, J. E. The effects of chemical interactions and culture history on the colonization of structured habitats by competing bacterial populations. *Bmc Microbiol* **14**, (2014).
- 6 Hays, S. G., Patrick, W. G., Ziesack, M., Oxman, N. & Silver, P. A. Better together: engineering and application of microbial symbioses. *Current Opinion in Biotechnology* **36**, 40-49, (2015).
- 7 Sackmann, E. K., Fulton, A. L. & Beebe, D. J. The present and future role of microfluidics in biomedical research. *Nature* **507**, 181-189, (2014).
- 8 Paratore, F., Bacheva, V., Bercovici, M. & Kaigala, G. V. Reconfigurable microfluidics. *Nat Rev Chem* **6**, 70-80, (2022).
- 9 de Jong, J., Lammertink, R. G. H. & Wessling, M. Membranes and microfluidics: a review. *Lab Chip Lab Chip* **6**, 1125-1139, (2006).
- 10 Trantidou, T., Friddin, M. S., Salehi-Reyhani, A., Ces, O. & Elani, Y. Droplet microfluidics for the construction of compartmentalised model membranes. *Lab Chip Lab Chip* **18**, 2488-2509, (2018).
- 11 Dai, Z. J., Lee, A. J., Roberts, S., Sysoeva, T. A., Huang, S. Q., Dzuricky, M., Yang, X. Y., Zhang, X., Liu, Z. H., Chilkoti, A. & You, L. C. Versatile biomanufacturing through stimulus-responsive cell-material feedback. *Nature Chemical Biology* **15**, 1017-+, (2019).
- 12 Ponomarova, O., Gabrielli, N., Sevin, D. C., Mulleder, M., Zimgibl, K., Bulyha, K., Andrejev, S., Kafkia, E., Typas, A., Sauer, U., Ralser, M. & Patil, K. R. Yeast Creates a Niche for Symbiotic Lactic Acid Bacteria through Nitrogen Overflow. *Cell Systems* **5**, 345-+, (2017).
- 13 Francis, D. M. & Page, R. Strategies to optimize protein expression in E. coli. *Curr Protoc Protein Sci* **Chapter 5**, Unit 5 24 21-29, (2010).
- 14 Satoh, S., Ikeuchi, M., Mimuro, M. & Tanaka, A. Chlorophyll b expressed in cyanobacteria functions as a light harvesting antenna in photosystem I through flexibility of the proteins. *Journal Of Biological Chemistry* **276**, 4293-4297, (2001).
- 15 Xu, H., Vavilin, D. & Vermaas, W. Chlorophyll b can serve as the major pigment in functional photosystem II complexes of cyanobacteria. *Proceedings Of the National Academy Of Sciences Of the United States Of America* **98**, 14168-14173, (2001).
- 16 Haupts, U., Maiti, S., Schwille, P. & Webb, W. W. Dynamics of fluorescence fluctuations in green fluorescent protein observed by fluorescence correlation spectroscopy. *Proc Natl Acad Sci U S A* **95**, 13573-13578, (1998).
- 17 Young, B., Wightman, R., Blanvillain, R., Purcel, S. B. & Gallois, P. pH-sensitivity of YFP provides an intracellular indicator of programmed cell death. *Plant methods* **6**, 27, (2010).
- 18 Cranfill, P. J., Sell, B. R., Baird, M. A., Allen, J. R., Lavagnino, Z., de Gruiter, H. M., Kremers, G. J., Davidson, M. W., Ustione, A. & Piston, D. W. Quantitative assessment of fluorescent proteins. *Nat Methods* **13**, 557-562, (2016).
- 19 Lau, P. S., Tam, N. F. Y. & Wong, Y. S. Wastewater nutrients (N and P) removal by carrageenan and alginate immobilized *Chlorella vulgaris*. *Environ Technol* **18**, 945-951, (1997).
- 20 Pierobon, S. C., Riordon, J., Nguyen, B., Ooms, M. D. & Sinton, D. Periodic Harvesting of Microalgae From Calcium Alginate Hydrogels for Sustained High-Density Production. *Biotechnology And Bioengineering* **114**, 2023-2031, (2017).

- 21 Pane, L., Feletti, M., Bertino, C. & Carli, A. Viability of the marine microalga *Tetraselmis suecica* grown free and immobilized in alginate beads. *Aquacult Int* **6**, 411-420, (1998).

Reviewers' Comments:

Reviewer #1:

Remarks to the Author:

The authors have adequately addressed my concerns in the revision.

Reviewer #3:

Remarks to the Author:

The authors have done a good and thorough job responding to reviewer comments to clarify the potential advantages to the MSB approach. In particular demonstrating the 73% higher yield in MSBs is impressive and appears to rely on artificially inducing various stresses via immobilization before global nutrients have been depleted, allowing for greater conversion to product yields. Future work should focus on identifying the metabolic mechanisms associated with this and which key applications would benefit from MSBs.

Reviewer #4:

Remarks to the Author:

The authors have addressed all my concerns. Suggest the acceptance.

Below, we provide point-by-point responses to reviewers' comments (*italicized, blue*). The revision in the manuscript is highlighted in blue.

Reviewer #1 (Remarks to the Author):

The authors have adequately addressed my concerns in the revision.

We thank the reviewer again for providing constructive comments.

Reviewer #3 (Remarks to the Author):

The authors have done a good and thorough job responding to reviewer comments to clarify the potential advantages to the MSB approach. In particular demonstrating the 73% higher yield in MSBs is impressive and appears to rely on artificially inducing various stresses via immobilization before global nutrients have been depleted, allowing for greater conversion to product yields. Future work should focus on identifying the metabolic mechanisms associated with this and which key applications would benefit from MSBs.

We thank the reviewer for the helpful and constructive comments. We will continue focusing on optimizing and expanding the technology and application of MSB and MSBC. We will also discover and identify the underlying metabolic mechanism of micro-organism inside MSB and MSBC to understand their unique behavior.

We thank the reviewer again for providing constructive comments.

Reviewer #4 (*Remarks to the Author*):

The authors have addressed all my concerns. Suggest the acceptance.

We thank the reviewer again for providing constructive comments.